# Toxoplasma gondii excretion of glycolytic products is associated with acidification of the parasitophorous vacuole during parasite egress

**My-Hang Huynh, Vern B. Carruthers** *

Department of Microbiology and Immunology, University of Michigan Medical School, Ann Arbor, Michigan, United States of America

* vcarruth@umich.edu

**Data Availability Statement:** All relevant data are within the manuscript and the Supporting Information.

## Abstract

The *Toxoplasma gondii* lytic cycle is a repetition of host cell invasion, replication, egress, and re-invasion into the next host cell. While the molecular players involved in egress have been studied in greater detail in recent years, the signals and pathways for triggering egress from the host cell have not been fully elucidated. A perforin-like protein, PLP1, has been shown to be necessary for permeabilizing the parasitophorous vacuole (PV) membrane or exit from the host cell. *In vitro* studies indicated that PLP1 is most active in acidic conditions, and indirect evidence using superecliptic pHluorin indicated that the PV pH drops prior to parasite egress. Using ratiometric pHluorin, a GFP variant that responds to changes in pH with changes in its bimodal excitation spectrum peaks, allowed us to directly measure the pH in the PV prior to and during egress by live-imaging microscopy. A statistically significant change was observed in PV pH during ionomycin or zaprinast induced egress in both wild-type RH and Δ*plp1* vacuoles compared to DMSO-treated vacuoles. Interestingly, if parasites are chemically paralyzed, a pH drop is still observed in RH but not in Δ*plp1* tachyzoites. This indicates that the pH drop is dependent on the presence of PLP1 or motility. Efforts to determine transporters, exchangers, or pumps that could contribute to the drop in PV pH identified two formate-nitrite transporters (FNTs). Auxin induced conditional knockdown and knockouts of FNT1 and FNT2 reduced the levels of lactate and pyruvate released by the parasites and lead to an abatement of vacuolar acidification. While additional transporters and molecules are undoubtedly involved, we provide evidence of a definitive reduction in vacuolar pH associated with induced and natural egress and characterize two transporters that contribute to the acidification.

## Author summary

*Toxoplasma gondii* is a single celled intracellular parasite that infects many different animals, and it is thought to infect up to one third of the human population. This parasite must rupture out of its replicative compartment and the host cell to spread from one cell

**Funding:** This work was supported by an operating grant from the National Institutes of Health Grant (R01AI046675 to V.B.C.). The funder had no role in study design, data collection and analysis, decision to publish, or preparation of the manuscript.

**Competing interests:** The authors have declared that no competing interests exist.

to another. Previous studies indicated that a decrease in pH occurs within the replicative compartment near the time of parasite exit from host cells, an event termed egress. However, it remained unknown whether the decrease in pH is directly tied to egress and, if so, what is responsible for the decrease in pH. Here we used a fluorescent reporter protein to directly measure pH within the replicative compartment during parasite egress. We found that pH decreases immediately prior to parasite egress and that this decrease is linked to parasite disruption of membranes. We also identified a family of transporters that release acidic products from parasite use of glucose for energy as contributing to the decrease in pH during egress. Our findings provide new insight that connects parasite glucose metabolism to acidification of its replicative compartment during egress from infected cells.

## Introduction

As an obligate intracellular pathogen, *Toxoplasma gondii* critically relies on efficiently completing each step of its lytic cycle for successful propagation and survival. In recent years, a greater focus has been applied to the egress step, wherein the parasites exit the host cell as a necessary prelude to infecting a neighboring cell. This increased attention has elucidated a well-orchestrated and complex cascade of events involving several molecular players, including activation of protein kinases by calcium ($Ca^{2+}$) or cyclic guanine monophosphate, release of proteins from apical microneme organelles, and activation of the glideosome motility machinery [1]. As with most other parasites belonging to the phylum Apicomplexa, *T. gondii* parasites replicate inside a membrane bound parasitophorous vacuole (PV), from which it must escape before leaving the host cell to initiate another round of the lytic cycle.

Most of the studies involving egress have been performed using chemical inducers such as ionomycin or zaprinast, which increase cytosolic $Ca^{2+}$ to elicit egress. A proposed general model for the activation of egress is as follows: (1) addition of inducer initiates parasite $Ca^{2+}$ signaling (reviewed in [2,3]); (2) host cytosolic $Ca^{2+}$ levels transiently increase by a mechanism that is not well understood [4]; (3) the parasite internalizes host-derived $Ca^{2+}$ through a nifedipine-sensitive $Ca^{2+}$ channel [4]; and (4) the internalized $Ca^{2+}$ increases parasite $Ca^{2+}$ levels to above a threshold [4], which activates the secretion of the micronemes, including the perforin-like protein 1 (PLP1), and facilitates rupture of the parasitophorous vacuole membrane (PVM) for egress. During spontaneous or natural egress, diacylglycerol kinase 2 (DGK2) was shown to be a plausible candidate for an intrinsic signal [5]. DGK2 is secreted into the PV and generates phosphatidic acid as a signaling molecule. DGK2-defective parasites were selectively defective in natural egress but were able to respond to chemical inducers. Conversely, egress is negatively regulated by the cAMP-dependent protein kinase A catalytic subunit 1 (PKAc1) through suppression of cyclic GMP (cGMP) cytosolic $Ca^{2+}$ signaling, as demonstrated by premature egress of PKAc1-deficient parasites [6,7]. These studies also implicated PV acidification as a determinant for the early egress of PKAc1-deficient tachyzoites based on treatment with a P-type ATPase inhibitor (dicyclohexylcarbodiimide, DCCD) [7] or neutralization with $NH_4Cl$ [6].

Early work showed that tachyzoite motility is pH-dependent, and that alkaline conditions inhibit motility whereas acidic buffers induce motility [8]. It was later shown that this pH-dependent motility is likely due to the effect of pH on microneme secretion, wherein low pH activates microneme secretion, and also leads to parasite egress [9]. Concomitantly, pH neutralization or treatment with DCCD was found to suppress tachyzoite egress. This study also noted that a drop in PV pH was associated with late-stage infection ~30 h post-inoculation, a decrease that was partially reversed upon treatment with the weak base $NH_4Cl$. The low pH

associated with egress was correlated with an increase in PLP1 activity. More specifically, in erythrocyte hemolysis assays, recombinant PLP1 lytic activity was shown to be most active between pH 5.4 and 6.4, a pH range for which PLP1 binding to erythrocyte ghost membranes was also enhanced [9]. *T. gondii* PLP1 is not unique in this respect, as several other pathogenic cytolytic proteins have been found to be more active at low pH, including *Listeria monocytogenes* listeriolysin O [10,11], *Leishmania amazonensis* a-leishporin [12], *Trypanosoma cruzi* TC-TOX [13], and *Bacillus thuringiensis* Cyt1A [14]. Also, acidification of transient PVs containing *Plasmodium yoelii* sporozoites was suggested to promote parasite escape in a PyPLP1-dependent manner [15]. For TgPLP1, increased activity at low pH is consistent with the observed lower pH of the PV beginning at ~30 h post-infection when parasites are beginning to egress [9]. However, the study was carried out using superecliptic pHluorin, which measures relative pH [16] rather than absolute pH. Also, the measurements were made on a population of infected cells over a relatively long period of time. Thus, this earlier work failed to capture changes in pH occurring during egress from individual cells.

Herein, we utilize a ratiometric pHluorin (RatpH) [16] to directly quantify PV pH in individual infected cells. Expression of RatpH in the PV consistently revealed a decrease in PV pH immediately preceding induced or natural egress. Using parasites lacking PLP1 we identified a role for this pore forming protein in PV acidification and $Ca^{2+}$ signaling within the host and parasite prior to PV rupture. We also identified formate-nitrite transporters (FNTs) as contributing to the acidification of the PV during egress, likely via co-transporting protons with the glycolytic products lactate and pyruvate. However, disruption of FNTs did not completely abrogate the release of lactate and pyruvate or eliminate the drop in PV pH, suggesting the involvement of other unidentified transporters or products that also contribute to PV acidification during egress.

## Results

### PV pH decreases immediately prior to induced and natural egress

Green fluorescent protein (GFP) has two excitation peaks at 410 nm and 470 nm. This property has been exploited to create, via amino acid substitutions, pH sensitive variants termed pHluorins [16]. Ratiometric pHluorin (RatpH) responds to a decrease in pH with reduced fluorescence from excitation at 410 nm and increased fluorescence from excitation at 470 nm. To measure the pH of the *T. gondii* PV, we expressed a codon-optimized RatpH in RH parasites (RH-RatpH). We then tested the ability of RatpH to respond to changes in pH by incubating RH-RatpH infected cells in buffers of defined pH containing nigericin to equilibrate $H^+$ across membranes (**Fig 1A**). Obtaining ratiometric images by excitation at 410 nm and 470 nm (**Fig 1B**) allowed the generation of a calibration curve for pH (**Fig 1C**). These findings show that RatpH is a robust indicator of pH when expressed in the PV and thus is suitable for measuring changes in PV pH during egress.

To determine if the expression of RatpH in RH affected parasite egress or fitness, we compared RH and RH-RatpH parasites for extent of egress and time to egress in response to ionomycin or zaprinast and their growth in a co-culture competition assay. We found no difference in the percentage of parasites that egressed within 5 min of induction (**S1A Fig**) or in the time to egress (**S1B Fig**). Furthermore, there was no difference in relative abundance of each strain following serial co-culture for 5 passages (**S1C Fig**).

To initially quantify changes in pH during egress we performed live ratiometric imaging of RH-RatpH infected cells in $Ca^{2+}$ containing buffer upon adding the $Ca^{2+}$ ionophore ionomycin, which induces egress by equilibrating calcium across membranes. Collecting a series of ratio images following ionomycin treatment showed that a decrease in PV pH occurs prior to

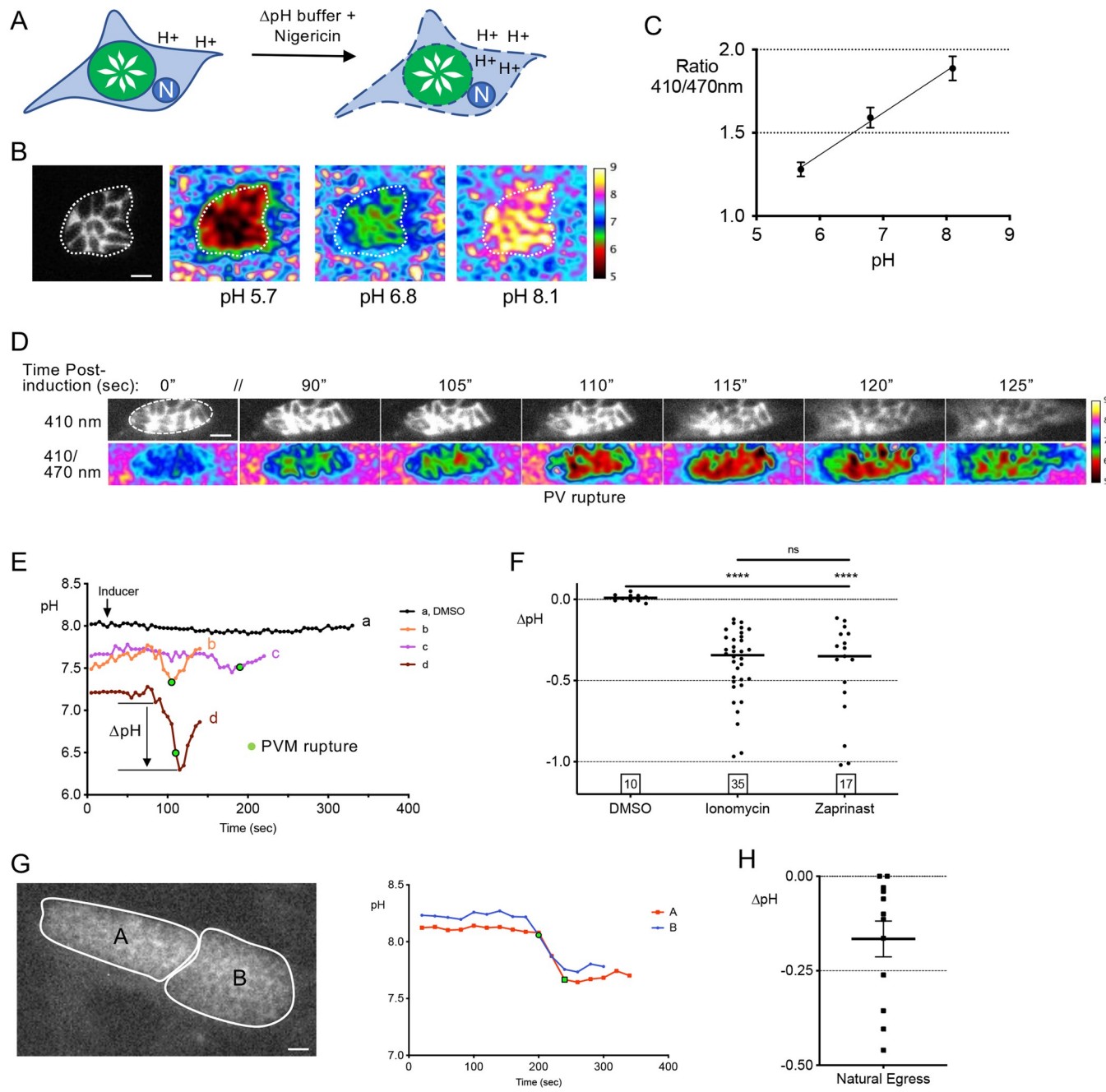

**Fig 1. Measurement of pH in *T. gondii* PVs expressing ratiometric pHluorin.** A) Equilibration of vacuoles to the surrounding buffer of known pH values with nigericin. B) Vacuole ratio images at 410/470 nm that are false colored according to the scale shown to the right. C) Calibration curve from the ratio 410/470 nm values for each pH value (mean ± S.D); $R^2 = 0.9992$. Scale bar, 5 μm. D) Reductions in PV pH during egress observed by live imaging of ionomycin-induced RH-RatpH parasites. The upper image series is from excitation at 410 nm to visualize the time of egress i.e., when pHluorin first leaves the PV due to rupture of the PVM. The lower series is of ratio images according to the pH scale bar shown on the right. // indicates a gap in time. E) The magnitude of pH changes varies between PVs. Tracings of pH values of RH-RatpH PVs induced with DMSO (a) or ionomycin (b, c, d). Tracing d represents the vacuole shown in 1D. Image acquisition was paused following frame 5 (25 sec), inducer was added, and acquisition was started again. Green data points indicate the time when pHuorin first leaves the PV, which is designated as the time of egress. F) A significant reduction in PV pH occurs after ionomycin or zaprinast induction of egress. Data points represent changes in PV pH starting from baseline to a drop greater than 0.05 following induction with either ionomycin or zaprinast. Numbers in squares within the graph indicates the numbers of PVs enumerated. A Kruskal-Wallis test with Dunn's multiple comparison was performed. **** $p \leq 0.0001$. Bars indicate the median. ns, not significant. G) A drop in PV pH occurs during natural egress. Representative PVs late in replication (~50 h) and pH tracings associated with egress. Scale bar, 10 μm. H) Quantification of pH changes in PVs during natural egress. Bar indicates the mean.

rupture of the PVM, which was indicated by the release of RatpH from the PV in non-ratioed images (**Fig 1D**). Tracings from individual PVs showed that in contrast to a consistent pH observed upon treatment with vehicle (DMSO) (**Fig 1E, a**), PV pH decreased 10–30 sec prior to PV rupture in infected cells treated with ionomycin (**Fig 1E, b,c,d and S1 Video**). To quantify the decrease in PV pH from the tracings, we took the lowest pH value, which typically occurred near the time of PV rupture (identified by pHluorin release) and subtracted it from the pH value that was the highest immediately prior to the descent, as described more extensively in the Materials and Methods. We found that although the magnitude of the decrease was variable, all the PVs we observed showed a drop in PV pH prior to PV rupture (**Fig 1F**). Analysis of 35 PVs recorded over multiple experiments showed a mean decrease of 0.35 pH units, corresponding to a ~3-fold increase in $[H^+]$ within the PV. We also observed a similar decrease in PV pH upon inducing egress with zaprinast (**Fig 1F**), which activates parasite protein kinase G upstream of $Ca^{2+}$ signaling [17], a mechanism distinct from that of ionomycin.

To determine if a decrease in pH occurs during natural egress, we imaged RH-RatpH infected cells at ~48–52 h post-infection for 20 min/field of view. We observed a decrease in PV pH in 10 out of 12 egress events, with a mean decrease of 0.17 pH units (**Fig 1G and 1H and S2 Video**). Taken together, our findings suggest that PV pH decreases immediately prior to induced and natural egress. Due to the challenges of capturing natural egress events, all subsequent experiments were performed with induced egress.

### Ionic environment modulates the vacuolar pH response

Earlier work suggested that a loss of cytoplasmic $K^+$ from the host cell triggers egress of *T. gondii* tachyzoites through activation of phospholipase C and a subsequent increase in cytoplasmic $Ca^{2+}$ [18]. A more recent study proposed that the decrease in $K^+$ accelerates egress but is not a trigger [4]. While $Ca^{2+}$ is necessary for activation of egress, the absence of extracellular $Ca^{2+}$ delayed but did not inhibit parasite egress [2]. pH measurements presented thus far have been obtained in Ringer's buffer (155 mM NaCl, 3 mM KCl, 1 mM $Ca^{2+}$, 1 mM $MgCl_2$, 10 mM glucose, 10 mM HEPES, 3 mM $NaH_2PO_4$, pH 7.4). To test the effect of $Ca^{2+}$ and $K^+$ in the extracellular media on the vacuolar pH drop, infected cells were incubated in Ringer's or 145 mM $K^+$ (termed high $K^+$) to mimic the $[K^+]$ of the host cell cytosol, each with and without $Ca^{2+}$. Buffer compositions used in this study are listed in **Table 1**.

**Table 1. Ion composition of buffers used in this study.**

|  | Ringer's | Ringer's-Ca$^{2+}$ | High K$^+$ | High K$^+$-Ca$^{2+}$ | EC-Na$^+$ | EC-Cl$^-$ |
|---|---|---|---|---|---|---|
| NaCl | 155 mM | 155 mM | 5 mM | 5 mM |  |  |
| KCl | 3 mM | 3 mM | 145 mM | 145 mM | 5.8 mM |  |
| CaCl$_2$ | 1 mM |  | 2 mM |  | 1 mM |  |
| MgCl$_2$ | 1 mM | 1 mM | 1 mM | 1 mM | 1 mM |  |
| Glucose | 10 mM | 10 mM | 10 mM | 10 mM | 10 mM |  |
| HEPES | 10 mM | 10 mM | 15 mM | 15 mM | 10 mM |  |
| NaH$_2$PO$_4$ | 3 mM | 3 mM |  |  |  |  |
| EGTA |  | 100 μM |  | 1 mM |  |  |
| Choline-Cl |  |  |  |  | 142 mM |  |
| Na$^+$ glutamate |  |  |  |  |  | 155 mM |
| K$^+$ glutamate |  |  |  |  |  | 3 mM |
| Ca$^{2+}$ glutamate |  |  |  |  |  | 2 mM |
| MgSO$_4$ |  |  |  |  |  | 1 mM |

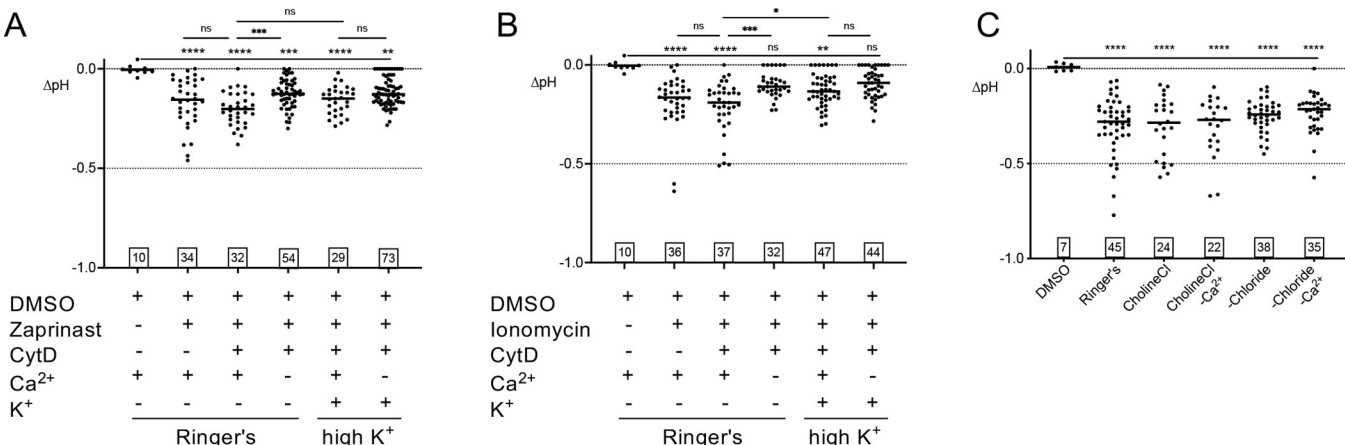

**Fig 2. Composition of surrounding buffer mildly abates PV pH changes.** A) and B) Extracellular buffer without $Ca^{2+}$ tempers the drop in pH with zaprinast (A) or ionomycin (B) induction. RH-RatpH PVs, treated with or without CytD, incubated in Ringer's buffer or High $K^+$ with or without $Ca^{2+}$ and induced with DMSO, ionomycin, or zaprinast. ns, not significant. C) $Na^+$ and $Cl^-$ do not affect pH changes. Extracellular buffer with $Na^+$ replaced with choline or all $Cl^-$ replaced with glutamate, with or without $Ca^{2+}$. Numbers in squares within all graphs indicate the numbers of vacuoles enumerated. A Kruskal-Wallis test with Dunn's multiple comparison was performed for all graphs. Bars indicate the median. * $p<0.05$, ** $p<0.01$, *** $p<0.001$, **** $p<0.0001$.

Egress from infected cells in Ringer's buffer was induced with zaprinast (**Fig 2A**) or ionomycin (**Fig 2B**) with or without cytochalasin D (CytD) treatment to determine whether immobilizing the parasites affects the decrease in PV pH after induction. Since CytD did not markedly affect PV pH, to facilitate imaging it was included in all subsequent experiments unless otherwise noted. Whereas removal of $Ca^{2+}$ from Ringer's significantly attenuated the drop in PV pH after induction with zaprinast or ionomycin, removal of $Ca^{2+}$ from the high $K^+$ had no effect. In the presence of $Ca^{2+}$, incubation in high $K^+$ showed a trend toward attenuating the drop in PV pH after induction with zaprinast and a significant attenuation following ionomycin induction. When $Na^+$ in the extracellular buffer is replaced with choline to maintain the total concentration of monovalent cations, with or without $Ca^{2+}$, there was no effect on the pH drop observed (**Fig 2C**). A role for chloride was assessed by replacing it with glutamate or sulfate (-Chloride buffer) composed of potassium glutamate, sodium glutamate, calcium glutamate, and magnesium sulfate. The absence of chloride also had no effect on PV acidification (**Fig 2C**). Together these findings imply that extracellular $Ca^{2+}$ is necessary for normal acidification of the PV during egress and that the loss of $K^+$ from host cells also influences PV pH.

## PLP1 influences acidification of the PV and calcium signaling in the parasite

Parasites deficient in PLP1 were previously shown to be substantially delayed in induced egress, with a proportion of the parasites unable to leave the vacuole [19]. Since PLP1 is a pore-forming protein, we reasoned that it could facilitate ion flux associated with acidification of the PV during egress. To address this, we introduced RatpH into PLP1 deficient parasites (Δ*plp1*-RatpH). As expected, Δ*plp1*-RatpH showed a delay in egress, with parasite egress occurring substantially later than the onset of motility (**Fig 3A**). We found that although Δ*plp1*-RatpH vacuoles from which the parasite was able to egress exhibited a drop in pH with ionomycin or zaprinast induction, this decrease was significantly attenuated relative to RH-RatpH (**Fig 3B**). Whereas the measurements thus far were performed on the entire PV, we noted that some regions of the PV show greater acidification. Thus, we also measured the

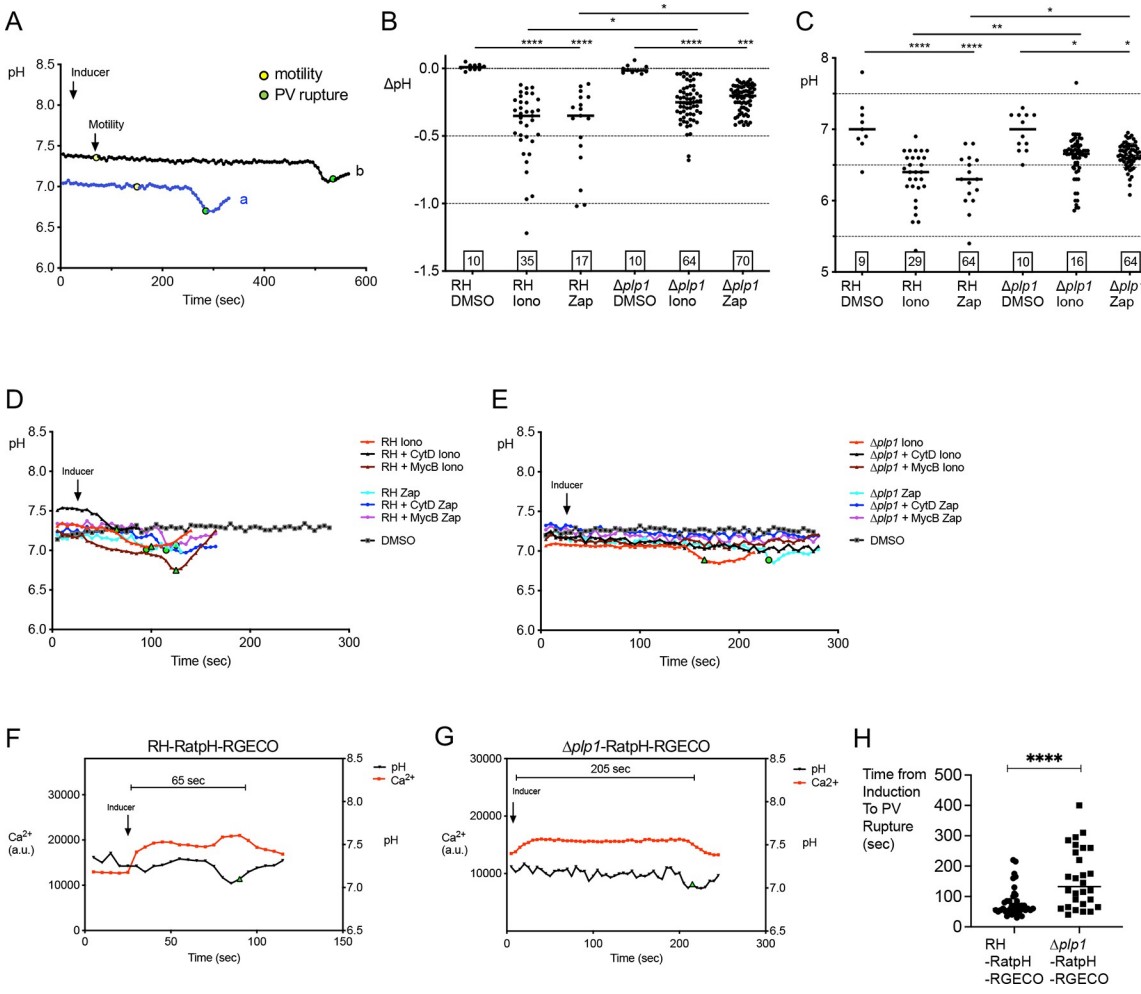

**Fig 3. PV acidification is attenuated in Δ*plp1* parasites.** A) Representative tracings of pH values in Δ*plp1*-RatpH from live imaging. Arrow indicates the time point that inducer was added. The yellow data points indicate the time point where motility of the tachyzoites began and the green points indicate the time of PV rupture. B) Δ*plp1* PVs show a significant attenuation of PV acidification after induction with ionomycin or zaprinast. Each point represents the change in pH within an individual PV. RH-RatpH data sets are the same as those presented in Fig 1F. Values in squares within the graph indicates the numbers of PVs enumerated. RH, RH-RatpH; Δ*plp1*, Δ*plp1*-RatpH. C) Punctate foci in individual vacuoles reach lower pH than whole vacuole. Ratio images of RH-RatpH or Δ*plp1*-RatpH vacuoles were analyzed in ImageJ using the Process Plugin to find min-max values (v.1.00). Data points indicate the lowest pH measured within a PV. RH, RH-RatpH; Δ*plp1*, Δ*plp1*-RatpH. Kruskal-Wallis tests with Dunn's multiple comparison were performed for B) and C). * *p*≤0.05, ** *p*≤0.01, **** *p*≤0.001, **** *p*≤0.0001. Bars indicate the median. D) Paralyzed WT parasites undergo PV acidification. Representative traces of RH-RatpH parasites untreated or immobilized with cytochalasin D (CytD) or mycalolide B (MycB), followed by ionomycin or zaprinast induction. RH, RH-RatpH; Δ*plp1*, Δ*plp1*-RatpH. E) Representative traces of Δ*plp1*-RatpH parasites untreated or immobilized with CytD or MycB, followed by ionomycin or zaprinast induction. F,G) Kinetics of parasite Ca²⁺ signaling relative to PV acidification. Representative traces of relative Ca²⁺ levels (left axis) and pH (right axis) of RH-RatpH-RGECO PVs (F) and Δ*plp1*-RatpH-RGECO PVs (G) following zaprinast induction. a.u., arbitrary units. H) Enumeration of 41 RH-RatpH-RGECO and 28 Δ*plp1*-RatpH-RGECO PVs shows an increase in the time from addition of inducer to PV rupture in Δ*plp1*-RatpH-RGECO. A two-tailed student's *t*-test was performed, **** *p*≤0.0001.

minimum pH in each PV prior to egress. This analysis showed that the mean difference in the lowest regional PV pH for non-induced (DMSO) and induced (zaprinast or ionomycin) RH-RatpH parasites was ~0.7 pH units, while for Δ*plp1*-RatpH parasites, it was ~0.4 pH units (**Fig 3C**). The analysis of lowest regional pH also confirmed an attenuation of PV acidification during induced egress of Δ*plp1*-RatpH parasites.

While *PLP1*-deficient parasites are defective in egress, some Δ*plp1* parasites are capable of egressing due to the motility of the parasites that eventually break through the PVM and

plasma membrane of host cells. To more selectively study the role of PLP1 and motility in the pH drop associated with egress, RH-RatpH or Δ*plp1*-RatpH parasites were paralyzed with either CytD or mycalolide B (MycB), which prevent actin polymerization or severs F-actin, respectively. As observed earlier in this study (**Fig 2A and 2B**), PVs of immobilized RH-RatpH parasites induced with ionomycin or zaprinast still displayed a significant drop in pH associated with pHluorin release (**Fig 3D**). However, PVs of immobilized Δ*plp1*-RatpH showed no decrease in pH and they failed to release pHluorin for the entire duration of the experiment (up to 20 min) in all 225 vacuoles observed (**Fig 3E**). Taken together, these findings suggest that PLP1 influences PV pH and that the absence of PLP1 or motility precludes PV acidification and rupture.

Since $Ca^{2+}$ signaling triggers the secretion of microneme proteins including PLP1 and it activates motility, we sought to define the timing of parasite cytosolic $Ca^{2+}$ and its relationship to changes in PV pH during egress by stably expressing a red genetically encoded $Ca^{2+}$ indicator RGECO in the cytosol of RH-RatpH and Δ*plp1*-RatpH parasites. $Ca^{2+}$ and PV pH measurements were obtained from motility competent parasites so that we could define the timing of $Ca^{2+}$ and pH dynamics in relation to PV rupture and egress. RH-RatpH-RGECO parasites displayed a rapid initial increase in cytosolic $Ca^{2+}$ after zaprinast induction and well before the drop in PV pH and egress (**Fig 3F and S3 and S4 Videos**). To determine if the order of signal acquisition during imaging (i.e., $Ca^{2+}$ then pH or pH then $Ca^{2+}$, at each time point) influenced the results, we compared $Ca^{2+}$ and pH dynamics by collecting data in both orders and found no difference (**S2 Fig**). The decrease in PV pH often corresponded to a second peak of $Ca^{2+}$, the presence of which has been reported previously as a potentiation of the initial peak via $Ca^{2+}$ influx from the host cell and the media [2]. Consistent with a prior report [4], Δ*plp1*-RatpH-RGECO parasites usually did not have a potentiation peak (**Fig 3G**). Also, whereas RH-RatpH-RGECO parasites ruptured the PVM 60 sec after induction on average, Δ*plp1*-RatpH-RGECO parasites took more than twice as long (average of 132 sec) to rupture the PVM (**Fig 3H**). From these findings we conclude that zaprinast induction initiates parasite $Ca^{2+}$ signaling followed by the potentiation of the $Ca^{2+}$ signal and PV acidification, and that the timing of PV acidification and rupture is delayed in parasites lacking PLP1.

## PLP1 contributes to the initiation of egress by dictating $Ca^{2+}$ influx into host cells

Previous studies have shown that a transient rise in host $Ca^{2+}$ occurs prior to parasite egress [4]. To confirm this elevation of host $Ca^{2+}$ and analyze its temporal relationship with $Ca^{2+}$ signaling in the parasite and acidification of the PV, we infected RGECO-expressing host cells with RH-RatpH-RGECO parasites, thereby allowing us to monitor host $Ca^{2+}$, parasite $Ca^{2+}$, and PV pH by imaging reporters that are spatially (host RGECO vs parasite RGECO) or spectrally (RGECO vs pHluorin) distinct. Upon zaprinast induction we observed an initial peak of parasite $Ca^{2+}$ like previous experiments, which was later followed by coincident elevation of host and parasite $Ca^{2+}$ and PV acidification prior to PV rupture (**Fig 4A and 4B**). To confirm this, we loaded infected host cells with the synthetic $Ca^{2+}$ indicator Cal-590-AM and again noted the coincident elevation of host $Ca^{2+}$ and PV acidification and rupture (**Fig 4C, left panel and S5 Video**). Our earlier results indicated that extracellular $Ca^{2+}$ in the medium is necessary for normal PV acidification. This prompted us to examine how the absence of extracellular $Ca^{2+}$ in the medium affects the elevation of host $Ca^{2+}$ during egress by adding zaprinast in $Ca^{2+}$ free medium. We found that the absence of extracellular $Ca^{2+}$ blunted the elevation of host $Ca^{2+}$ (**Fig 4C, right panel**), suggesting that the elevation of host $Ca^{2+}$ is due to influx of $Ca^{2+}$ from the medium.

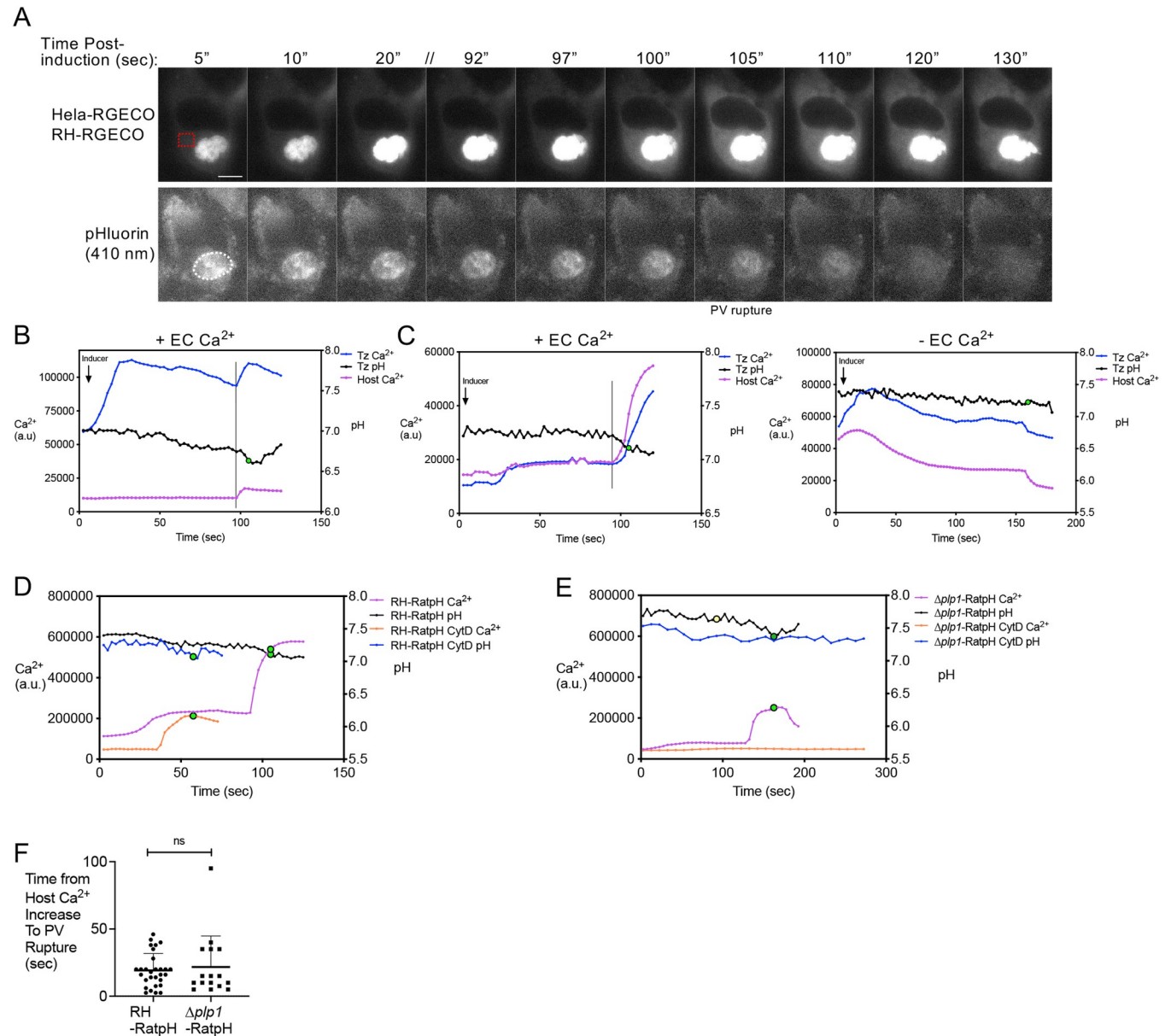

**Fig 4. Host cytosolic calcium increases prior to PV rupture.** A) Still images of time-course of HeLa-RGECO host $Ca^{2+}$ levels and PV rupture (loss of pHluorin from PV) following zaprinast induction. // indicates a gap in time. Scale bar, 10 μm. Dashed red outline indicates the region of the host cell used to analyze the RGECO fluorescence in B), and the dashed white outline indicates the PV used to analyze pH (pHluorin) and tachyzoite $Ca^{2+}$ (RGECO). B) Tracings of host and tachyzoite $Ca^{2+}$ levels (left axis) and PV pH (right axis) of vacuole shown in A. A vertical line denotes the time point after which host $Ca^{2+}$ levels increase and near when the PV pH decreases. Blue trace, parasite $Ca^{2+}$; black trace, PV pH; purple trace, host cell $Ca^{2+}$. a.u., arbitrary units. C) Representative tracings of an HFF host cell preloaded with Cal-590-AM, infected with RH-RatpH-RGECO parasites, and induced with zaprinast in Ringer's buffer with or without $Ca^{2+}$. A vertical line denotes the time point after which host $Ca^{2+}$ levels increase and when the PV pH decreases. D,E) Cal-590-AM-loaded HFF cells infected with RH-RatpH (D) or Δ*plp1*-RatpH (E), following treatment without or with CytD. Measurements of host $Ca^{2+}$ (left axis) and PV pH (right axis). Purple trace, host $Ca^{2+}$; black trace, PV pH; orange trace, host $Ca^{2+}$ with CytD-treated parasite, blue trace, PV pH with CytD-treated parasite. Yellow data point indicates the start of motility (in E) and green data points indicate PV rupture (in D and E). F) Time from host $Ca^{2+}$ (Cal-590 fluorescence) increase to PV pHuorin release in host cells infected with RH-RatpH (29 vacuoles) or Δ*plp1*-RatpH (16 vacuoles). A two-tailed student's *t*-test was performed. ns, not significant.

Previous work noted that PLP1 deficient parasites show muted $Ca^{2+}$ signaling and no elevation of host cell $Ca^{2+}$ when they attempt to egress naturally (i.e., after removing an inhibitory treatment) unless they ruptured the cell via motility [4]. These findings suggest that during

natural egress, PLP1 or motility is required for $Ca^{2+}$ signaling in both the parasite and host cell. However, the study did not test immobilized parasites to determine if PLP1 is necessary for host $Ca^{2+}$ signaling in the absence of motility-dependent PV rupture and egress. To determine if PLP1 and/or motility are required for host $Ca^{2+}$ signaling and when such signaling occurs relative to PV rupture, we measured host $Ca^{2+}$ with Cal-590-AM in cells infected with RH-RatpH without or with CytD treatment or with Δ*plp1*-RatpH without or with CytD treatment. As noted earlier, cells infected with motility competent RH-RatpH showed an elevation of host $Ca^{2+}$ prior to PV rupture, as did cells infected with CytD immobilized RH-RatpH (**Fig 4D**). This indicates that when PLP1 is present, motility is not necessary for the elevation of host $Ca^{2+}$ prior to PV rupture. Cells infected with motility competent Δ*plp1*-RatpH also showed an elevation of host $Ca^{2+}$ before PV rupture (**Fig 4E**). Notably though, elevation of host $Ca^{2+}$ occurred 15–30 sec prior to PV rupture, and motility initiated ~30–60 sec before the elevation of host $Ca^{2+}$. Taken together, this suggests that in the absence of PLP1, motility can stimulate elevation of host $Ca^{2+}$ before PV rupture and egress. Remarkably, cells infected with immobilized Δ*plp1*-RatpH parasites showed no host $Ca^{2+}$ signaling (12 infected host cells examined) (**Fig 4E**). Since the presence or absence of PLP1 is the only variable in this comparison, these findings indicate that PLP1 is necessary for host $Ca^{2+}$ signaling. Also, elevation of host $Ca^{2+}$ always preceded PV rupture (**Fig 4F**), suggesting that parasite motility or secretion of PLP1 triggers an elevation of host $Ca^{2+}$ before rupture of the PV. Taken together with the requirement for extracellular $Ca^{2+}$, our findings indicate that PLP1 (or motility) plays an early role in egress by directing the influx of $Ca^{2+}$ into infected host cells. Because the uptake of host-derived $Ca^{2+}$ by the parasite potentiates parasite $Ca^{2+}$ signaling [2], and PV acidification coincides with such potentiation, PLP1-dependent $Ca^{2+}$ signaling appears to be linked to PV acidification.

## The PV and host cytosol acidify simultaneously

Although the PVM contains pores that allow the diffusion of small molecules <1,300 Da including ions [2,20,21], we nevertheless attempted to determine the directionality of acidification by transiently expressing ratiometric pHluorin in the cytosol of HeLa cells followed by infection with RH-RatpH parasites (**S3A and S3B Fig**). Ratio images following induction with ionomycin in the provided example and others seem to indicate a pH change first occurring in the PV, followed by acidification in the host cell (**S3C Fig**). However, pH measurements of the entire PV and a section of the host cytosol show an indistinguishable pattern of the pH changes (blue and green tracings in **S3D Fig**). Thus, we conclude that a distinction between the drop occurring first in either the PV or host cannot be made. Nevertheless, from these experiments we ruled out that ionomycin or zaprinast treatment directly affects the cytosolic pH of host cells by treating and analyzing uninfected cells (**S3D and S3E Fig**).

## Several plasma membrane proton transporters are not required for acidification of the PV

Next, we sought to identify the basis for acidification of the PV by initially focusing on proton transporters with the potential to be expressed on the parasite surface. A V-type $H^+$ ATPase (V-ATPase) is localized on the plasma membrane as well as to the plant-like vacuole (PLV)/ Vacuolar compartment (VAC, used hereafter) organelle. The V-ATPase was shown to function in maintaining cytoplasmic pH and the acidic pH of the VAC and immature rhoptries [22]. To test if the V-ATPase contributes to acidification of the PV, we transiently expressed RatpH in the inducible knockdown parasite strain of VHA1 (iVHA), a critical subunit of V-ATPase, generated in Stasic et al. [22]. We confirmed down-regulation of VHA1 protein

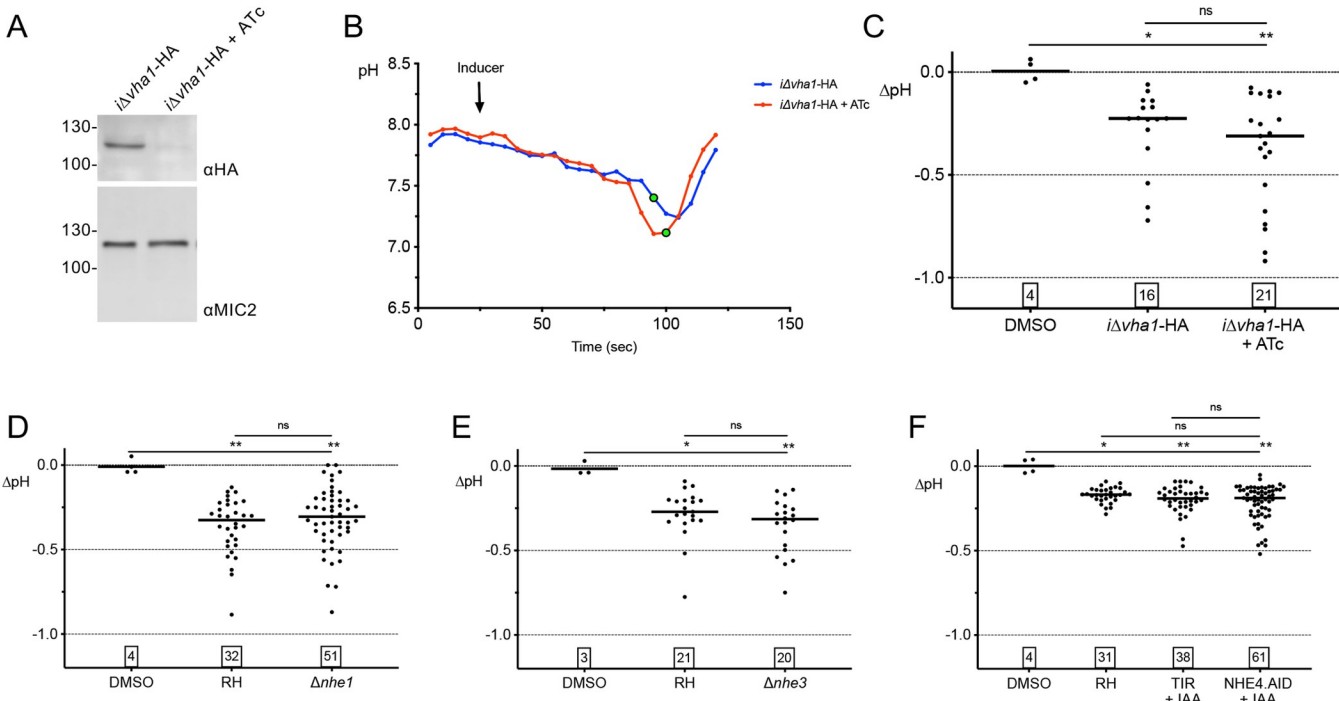

**Fig 5. Known plasma membrane-localized V-ATPase and sodium-hydrogen exchangers do not affect vacuolar pH changes during egress.** A) Western blot of inducible VHA1 (*iΔvha1*-HA) +/- anhydrotetracycline (ATc) treatment for 32 h. Blots were probed with Ms anti-HA to detect VHA1-HA; Rb anti-MIC2 was a loading control. B) Two representative pH tracings from C) of *iΔvha1*-HA parasites transiently transfected with RatpH, incubated +/−ATc, and induced with ionomycin. The green data points indicate the time point that pHluorin leaves the vacuole. C) Knockdown of *iΔvha1*-HA does not affect the magnitude of a pH change. ns, not significant. D-F) Sodium-proton exchangers do not contribute to vacuolar pH changes. Parasite strains Δ*nhe1* (D), Δ*nhe3* (E), and NHE4. AID + IAA (F) transiently transfected with RatpH were induced with ionomycin and pH changes measured by live-imaging. Numbers in squares within graphs indicate the numbers of vacuoles enumerated. Kruskal-Wallis tests with Dunn's multiple comparison was performed for all graphs. $^*$ $p \leq 0.05$, $^{**}$ $p \leq 0.01$. ns, not significant. Bars indicate the median.

levels following ATc treatment by western blotting for the HA tag appended to VHA1 (**Fig 5A**). We found that parasites lacking VHA1 (iΔ*vha1*-HA +ATc) showed a drop in PV pH after egress induction that was indistinguishable from those expressing VHA1 (iΔ*vha1*-HA) (**Fig 5B and 5C**). These findings suggest that VHA1 does not play a role in PV acidification during egress.

*T. gondii* expresses four sodium/proton exchangers: NHE1 is on the plasma membrane [23], NHE2 is associated with the rhoptries [24], NHE3 is associated with the VAC [25], and NHE4 is predicted to be on the plasma membrane or Golgi, depending on the prediction program used for the analysis [26]. Since it was unlikely that a rhoptry proton exchanger would affect PV acidification during egress, NHE2 was excluded from testing. We tested available knockout strains that lack NHE1 [23] or NHE3, [25], which we confirmed by PCR (**S4 Fig**). Upon transiently expressing RatpH, we found that Δ*nhe1* and Δ*nhe3* both showed normal acidification of the PV during induced egress (**Fig 5D and 5E**). To assess the potential contribution of NHE4 to pH changes in the PV during egress, we used CRISPR-Cas9 to append an auxin-inducible degron (AID) to the C-terminus of NHE4 (NHE4.AID) in RHΔ*ku80* parasites expressing the auxin receptor (TIR1) for AID-based protein degradation (RHΔ*ku80*/TIR, TIR hereafter) [27]. Immunofluorescence localization of the HA tag downstream of the AID indicated that the tagged NHE4 was found primarily in the region anterior to the nucleus, reminiscent of Golgi staining (**S5A Fig**). Addition of auxin (indoleacetic acid, IAA) effectively reduced expression of NHE4.AID to a level below detection based on western blotting

(**S5B Fig**). NHE4.AID parasites showed an acidification of the PV that was indistinguishable from that of RH or TIR parasites (**Fig 5F**). Altogether, this showed that NHE1, NHE3, and NHE4 do not contribute to the vacuolar acidification observed during egress.

## FNT1 and FNT2 are important for acidification of the PV during egress

Formate-nitrite transporters (FNTs) are multi-pass transmembrane proteins that transport monocarboxylate metabolites, such as formate and lactate, in *P. falciparum* [28,29] and in *T. gondii* [30]. *T. gondii* expresses three FNTs (FNT1-3) that localize to the parasite plasma membrane upon overexpression in tachyzoites [30]. Proteomics and transcriptomics data available in Toxodb suggest that FNT1 and FNT2 proteins are primarily expressed in tachyzoites whereas FNT3 is mainly expressed in the feline enteroepithelial stages. A previous study determined that TgFNTs transport L-lactate and formate in a pH-dependent manner, and it described small molecule inhibitors of FNT1-3 that impaired the tachyzoite lytic cycle [30]. More recent gene knockout studies reported that FNT1 plays a prominent role in lactate export [31,32].

To initially assess whether FNTs had a role in PV acidification, the two most potent inhibitors of parasite growth from the Erler et al study, BH-296 and BH-388 [30], were incubated with infected cells and PV pH changes during time-lapse microscopy were determined in RH-RatpH vacuoles. At 10 μM, BH-296 significantly reduced the magnitude of pH change during zaprinast induced egress compared to Ringer's buffer alone, whereas BH-388 had no effect, even at 10 μM (**Fig 6A**).

Since the inhibitors could target one or all the FNTs in *T. gondii* as well as potentially in the host cell, we created transgenic parasites to identify the contribution of FNT1 and FNT2 individually and together to the pH drop during egress; FNT3 was not initially examined given its lack of expression in the tachyzoite stage. A schematic of the strategy and the strains generated is illustrated in **Fig 6B** and **S6A Fig**. The mini auxin induced degron (mAID) cassette, including an mNeonGreen (mNG) and a Ty epitope tag (mNG-AIDTy), was introduced into the C-terminus of FNT1 by CRISPR-Cas9. Live imaging of the mNG tag as well as immunofluorescence assays of the Ty tag indicated the proper targeting and localization of FNT1-AIDTy (**Fig 6C** and **S7A Fig**) and correct tagging was confirmed by PCR (**S6B Fig**) and western blot (**Fig 6Di**). The mNG-AIDTy cassette was also introduced into the C-terminus of FNT2 (*iFNT2*) and confirmed by western blot (**Fig 6Dii**) but the mNG signal could not be detected by live imaging. The FNT1-AIDTy strain (*iFNT1*) was then used as the background in which FNT2 was tagged at the C-terminus with a 6xHA tag and HXGPRT selectable marker, generating the strain *FNT1*-AIDTy/*FNT2*-HA (*FNT2*-HA). Detection of the 6xHA could not be observed by immunofluorescence, similar to the inability to detect the Ty tag in *iFNT2*, perhaps due to low expression of FNT2. However, PCR (**S6B Fig**) and western blotting (**Fig 6Dii**) confirmed the integration and expression of the HA tag, respectively. The *FNT2* coding region was deleted using gRNAs targeting the 5' end of *FNT2* and the 3' end of the HXGPRT selectable cassette by replacement with the DHFR-TS selectable marker, generating the *iFNT1Δfnt2* strain (**S6A Fig**). Integration of the DHFR-TS in the FNT2 locus was confirmed by PCR (**S6B Fig**). The effectiveness of FNT1 knockdown with IAA was determined by both live imaging of the mNG tag (**S7A Fig**) and by western blotting for the Ty tag (**Fig 6Di** and **S7B Fig**). FNT1-AIDTy was essentially undetectable after 6 h of IAA treatment. Transient transfection of *iFNT1* parasites with RatpH and incubation with IAA for 20 h was performed prior to time-lapse imaging. We found that although inducible knockdown of FNT1 (*iFNT1* + IAA) or FNT2 (*iFNT2* + IAA) did not affect PV pH during egress, inducible knockdown of FNT1 in the absence of FNT2 (*iFNT1Δfnt2* + IAA) significantly attenuated the decrease in PV pH during egress, albeit

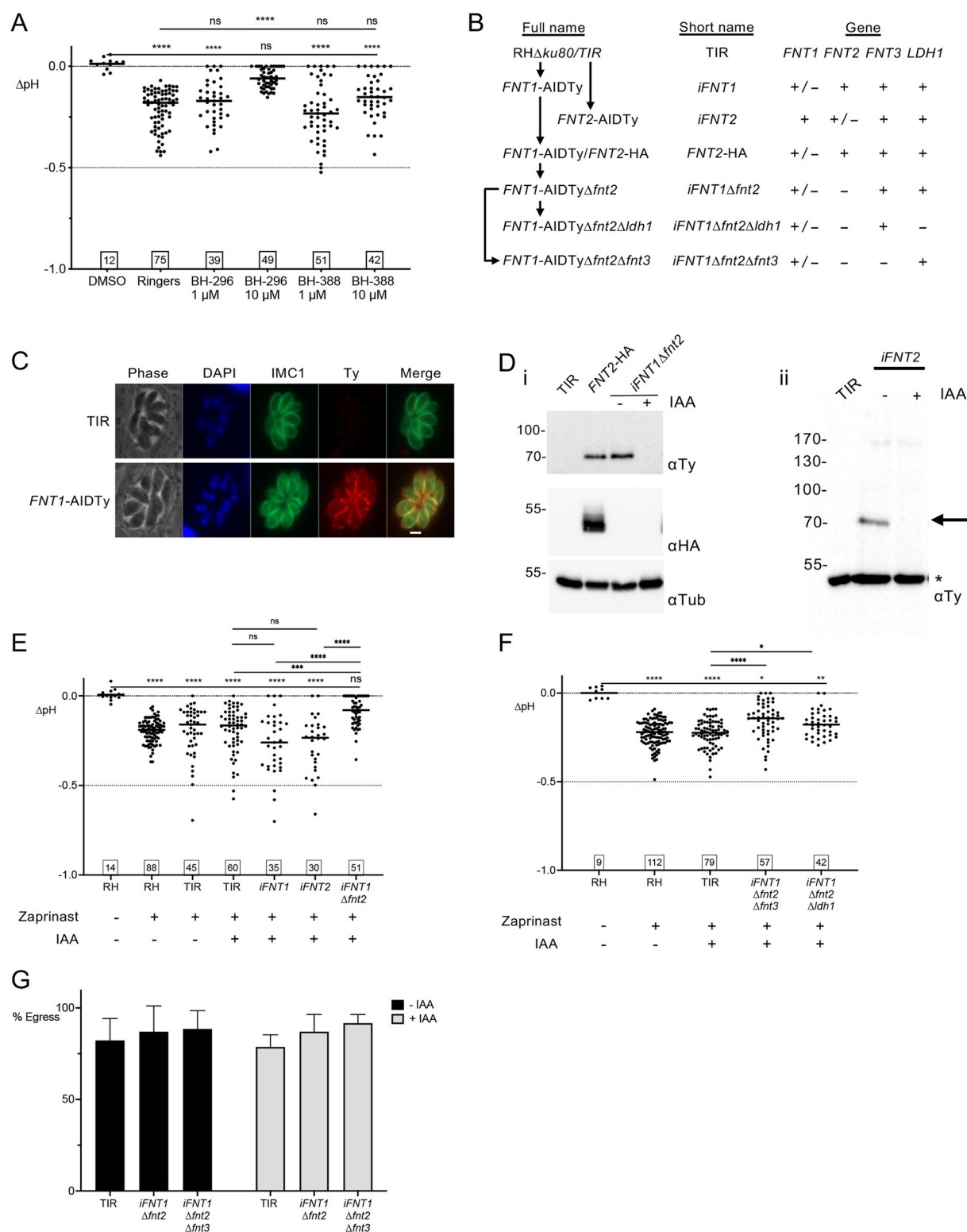

**Fig 6. Impairing formate-nitrite transporters attenuates PV acidification.** A) Effect of FNT inhibitors BH-296 and BH-388 at 1 μM or 10 μM on pH changes during zaprinast induced egress. B) Schematic description of FNT strains generated, with full and short names used in graphs. C) Immunofluorescence staining of parental TIR and FNT1-AIDTy intracellular vacuoles with anti-IMC1 and anti-Ty antibodies. Scale bar, 5 μm. D) IAA-treatment down-regulates FNT1 and FNT2 expression. Parasites treated for 24 h with or without IAA were purified and lysates were immunoblotted with anti-Ty; anti-HA showed expression of FNT2-HA. * indicates a non-specific band recognized by the anti-Ty

Ab. E) Disruption of FNT1 and FNT2 reduces the magnitude of PV acidification. ns, not significant. F) Knockout of *FNT3* or *LDH1* does not further attenuate the pH drop. G) Static egress assays of 28 h vacuoles of parental TIR and i*FNT1Δfnt2Δfnt3* ± IAA treatment. Error bars indicate the mean ± S.E.M. At least 250 PVs were enumerated. Numbers in squares within graphs indicate the numbers of vacuoles enumerated. Kruskal-Wallis tests with Dunn's multiple comparison was performed for all graphs. * $p \leq 0.05$, ** $p \leq 0.01$, **** $p \leq 0.001$, **** $p \leq 0.0001$. Bars indicate the median.

incompletely (**Fig 6E**). The PVs of some *iFNT1Δfnt2* + IAA parasites showed no detectable decrease in PV pH during egress, suggesting that PV acidification is not critical for parasite exit from host cells. *iFNT1* and *iFNT2* parasites without IAA treatment could not be analyzed by live imaging due to spectral overlap of mNG with RatpH. We also confirmed that there was no compensatory upregulation of FNT2 expression in *FNT2*-HA parasites following FNT1 depletion (**S7C Fig**). These findings suggest that FNT1 and FNT2 are involved in the observed pH drop in the PV during induced egress. Additionally, the magnitude of the pH drop in these strains being comparable to those observed for the inhibitors is supportive of the inhibitors targeting the parasites specifically versus the host FNTs.

## FNT3 does not play a role in PV acidification

Although FNT3 appears to be expressed almost exclusively in the sporozoite stages of the parasite based on data in Toxodb, compensation of FNT1 and FNT2 transport by upregulation of FNT3 expression in the absence of FNT1 and FNT2 was a possibility. To measure potential changes in FNT3 expression by reverse transcriptase quantitative PCR, we tested two independent primer sets to the FNT3 mRNA and used actin as the housekeeping gene for normalization. No significant change in FNT3 transcript level was observed in the absence of FNT1 and FNT2 (**S8 Fig**). To more conclusively rule out a role for FNT3 in the residual decrease of PV pH observed in *iFNT1Δfnt2*, we deleted FNT3 to generate an *iFNT1Δfnt2Δfnt3* strain (**S6A Fig**). We found that deletion of FNT3 in the absence of FNT1 and FNT2 did not further exacerbate the attenuation of PV acidification during egress, and that a residual drop of PV pH is still observed (**Fig 6F**). Despite the attenuation of a pH change in the PV during egress, neither the *iFNT1Δfnt2* nor the *iFNT1Δfnt2Δfnt3* parasites were defective in their ability to egress upon induction with zaprinast in a static (single 2 min time point) egress assay (**Fig 6G**). The absence of these FNTs also had no effect on intracellular replication (**S9 Fig**). Taken together our findings suggest that FNT1 and FNT2 contribute to acidification of the PV during egress, acidification is not critical for induced egress, and other transporters likely exist for residual acidification of the PV in parasites lacking the FNTs.

## Release of lactic acid and pyruvic acid potentially contribute to acidification of the PV

L-lactate is a major by-product of glycolysis. FNT1 was recently shown to be the principal lactate transporter in *T. gondii* tachyzoites [32]. Lactate dehydrogenases (LDHs) convert the end-product of glycolysis, pyruvate, to lactate. *T. gondii* harbors two LDHs, LDH1 and LDH2, with LDH1 being predominantly expressed in tachyzoites and LDH2 being expressed exclusively in the bradyzoite stage [33,34]. Pomel and colleagues [35] reported that several glycolytic enzymes including LDH1 relocalize from the cytosol to the parasite periphery during the transition from intracellular to extracellular parasites. The authors suggested this relocation could underlie subpellicular production of ATP from glycolysis for actinomyosin-dependent gliding motility during and following egress. However, several studies have reported that although *Δldh1* parasites are virulence attenuated and have defects during the chronic stage, they are not defective in the lytic cycle *in vitro* [33,36,37]. To determine whether the generation of

lactate by LDH1 contributed to the residual pH drop observed in the *iFNT1Δfnt2* strain, we disrupted *LDH1* by replacement with the bleomycin resistance cassette (**S6A Fig**). The resulting *iFNT1Δfnt2Δldh1* strain was transiently transfected with RatpH and pH changes during induced egress were measured. The loss of LDH1 in addition to FNT1 and FNT2 did not further attenuate the pH drop beyond that observed for *iFNT1Δfnt2* alone (**Fig 6F**), indicating that, while lactate release into the PV could contribute to the pH drop during egress, other molecules also play a role in this function.

The findings from *iFNT1Δfnt2Δldh1* parasites imply that lactate production is not essential for the observed pH drop during egress. To determine whether the release of lactate from the parasites could contribute to the changes in PV pH, we exposed extracellular TIR and FNT transgenic parasites (pretreated without or with IAA) to zaprinast (or DMSO) for 10 min at 37°C. Thereafter we pelleted the parasites and collected supernatants for analysis of released glycolytic products, akin to a previous study Kloehn et al. [31]. For the TIR parental strain, we found that zaprinast treatment showed a trend toward increasing the release of lactate, but the difference was not statistically significant (**Fig 7A**). Unexpectedly, *iFNT1* and *iFNT2* without IAA treatment both showed a significant reduction in lactate release relative to the parental strain (**Fig 7B**), implying that the mNG-AIDTy tag interferes with the function of these two transporters. That these strains showed a significant reduction in lactate release without an effect on PV pH indicates either that the level of lactate released is sufficient for the observed pH changes or other molecules are involved. The observed effects in *iFNT1* and *iFNT2* are inconsistent with the Zeng et al study [32], which suggested that FNT2 does not contribute significantly to the transport of L-lactate across the plasma membrane. Treatment with IAA did not further diminish the release of lactate, possibly due to the presence of the other transporter (FNT2 in *iFNT1* and FNT1 in *iFNT2*). Double (*iFNT1Δfnt2*) and triple knockout (*iFNT1Δfnt2Δfnt3)* strains appear to release even less lactate, which is consistent with a role for FNT1 and FNT2 in acidification of the PV during egress. As expected, lactate release by *iFNT1Δfnt2Δldh1* parasites was near the limit of detection. We conclude from these findings that release of lactate potentially contributes to the observed drop in PV pH during egress.

A similar trend of pyruvate release was observed in the strains and treatments. Interestingly, we found that zaprinast treatment increased the release of pyruvate from the TIR parental strain by 2- to 3-fold (**Fig 7C**), suggesting a possible link between $Ca^{2+}$ signaling and glycolysis. The relative release of pyruvate from TIR and FNT transgenic strains largely mirrored that of lactate (**Fig 7D**), consistent with the mNG-AIDTy tag impairing FNT1 and FNT2 transport function. Knocking down FNT1 in the absence of FNT2 (*iFNT1Δfnt2*), FNT2 and FNT3 (*iFNT1Δfnt2Δfnt3*) or FNT2 and LDH1 (*iFNT1Δfnt2Δldh1*) resulted in a more pronounced decrease in pyruvate export than that of lactate, thus confirming a role for FNT1 in pyruvate export. We observed no difference in pyruvate release between *iFNT1Δfnt2* and *iFNT1Δfnt2Δfnt3*, as expected given the lack of FNT3 expression in tachyzoites. Parasites lacking all three FNTs (*iFNT1Δfnt2Δfnt3*+IAA) still showed residual release of pyruvate, implying the existence of other unidentified transporters capable of exporting pyruvate. It is also possible that residual expression of FNT1 following IAA knockdown contributes to the release of pyruvate. Taken together, our findings imply an association of $Ca^{2+}$ signaling with glycolysis, roles for FNT1 and FNT2 in the export of lactate and pyruvate, and residual release of lactate and pyruvate by non-FNT transporters.

## Discussion

*T. gondii* egress from host cells is a tightly regulated and elaborate process with many upstream components, such as those that control microneme secretion including protein kinase G and

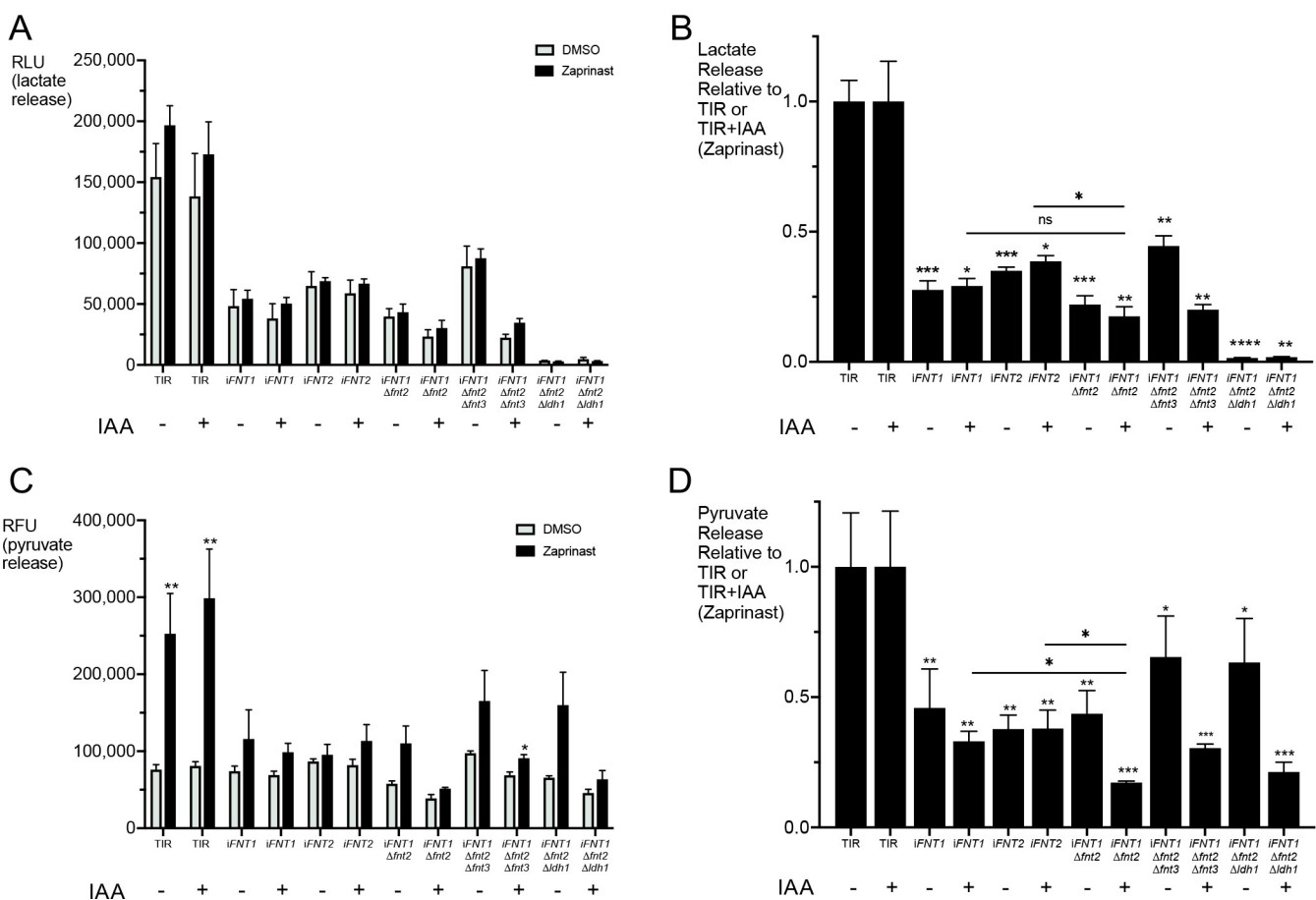

**Fig 7. Release of lactate and pyruvate could contribute to PV pH changes during egress.** A) Lactate release detected in supernatants following DMSO (vehicle control) or zaprinast induction. RLU, relative luminescence units. B) Lactate release expressed relative to parental TIR parasites ± IAA. ns, not significant. C) Pyruvate release detected in supernatants following DMSO (vehicle control) or zaprinast induction. RFU, relative fluorescence units. D) Pyruvate release expressed relative to parental TIR parasites ± IAA. Two-tailed student's $t$-test * $p \leq 0.05$, ** $p \leq 0.01$, **** $p \leq 0.001$, **** $p \leq 0.0001$. Error bars are mean ± S.E.M. Data in A)–D) represent 3 biological replicates with duplicates within each experiment.

$Ca^{2+}$-dependent protein kinases, downstream players in the gliding motility machinery, and cell-to-cell communication that coordinates the disassembly of the basal F-actin to disconnect parasites from one another (reviewed in [1]). The parasite also secretes several proteins that contribute to egress including perforin-like protein 1 (PLP1), which is released from the micronemes [19]. It was previously shown by Roiko et al [9] that PLP1 cytolytic and membrane binding activities are enhanced at low pH (5.4 to 6.9) compared to neutral pH (7.4). This study also confirmed an earlier report that low pH activates tachyzoite motility [8], and it extended insight by showing this occurs in part through the activation of microneme secretion, which is also necessary for motility. PLP1 activity and parasite motility both contribute to egress since the absence of one or the other delays egress but does not completely impair it [19,38]. The relationship between PLP1 activity and gliding motility at low pH was further linked when it was observed that a pH shift occurs during both induced egress and natural egress on a population scale [9]. Our expression of ratiometric pHluorin in the PV allowed for the direct measurement of pH changes within an individual vacuole during induced egress. The data definitively showed that a drop in pH occurs during both induced and natural egress.

The extent to which PV acidification facilitates egress remains unclear. On one hand, the mean regional minimum PV pH after induction (~6.3) is within the range of that for which PLP1 membrane binding and activity is enhanced (5.4–6.9) [9]. Although the relationship between environmental pH and motility has not been defined extensively, parasite exposure to pH 5.4 [9] or 6.0 to 7.3 [8] is known to activate motility. These findings are consistent with PV acidification augmenting PLP1 activity and parasite motility. On the other hand, the decrease in pH prior to egress showed substantial variation between PVs, and in some cases parasites egressed from host cells without a discernible change in PV pH. Attempts to measure the effect of disrupting PV acidification using the weak base $NH_4Cl$ or the P-type ATPase inhibitor DCCD were ineffective in our live-imaging set-up, despite their reported efficacy in static egress assays [9]. This discrepancy could be due to differences in the experimental set up or to the continuous exposure of the parasites to intense illumination during image acquisition, amongst other variables. Also, our inability to completely abrogate PV acidification by disrupting substrate transporters further limited defining the influence of PV pH on egress. Nevertheless, since we observe egress in instances where no acidification of the PV occurs (particularly with inhibitors or mutants), our findings suggest that PV acidification is not essential for parasite egress, which is consistent with egress being regulated at multiple levels to ensure successful liberation.

The extracellular ionic environment has been proposed to affect the initiation and progression of egress, particularly the reduction of $[K^+]$ following the breakdown of the host plasma membrane [18] and the presence of environmental $Ca^{2+}$ [2]. We found that high extracellular (EC) $[K^+]$ or the absence of EC $Ca^{2+}$ moderately attenuated PV acidification, suggesting that ion flux influences PV pH during egress. Recent studies measuring parasite cytosolic $Ca^{2+}$ during egress reported that WT parasites often show two peaks of $Ca^{2+}$, with the second peak being dependent on influx of EC $Ca^{2+}$ into the parasite from the environment [2,4]. Parasite uptake of $Ca^{2+}$ was sensitive to inhibition by nifedipine [4], suggesting the involvement of a voltage operated $Ca^{2+}$ channel in the plasma membrane of the parasite. This same study showed that parasites lacking PLP1 have aberrant and prolonged $Ca^{2+}$ signaling, which is generally consistent with our results, indicating muted signal potentiation from diminished parasite uptake of $Ca^{2+}$. We also found that PV acidification is not dependent on motility when PLP1 is present and that it is partially dependent on PLP1 when motility is functional; however, the absence of PLP1 and motility completely abrogates host cytosolic $Ca^{2+}$ influx, PV acidification and egress. Our data supports the hypothesis that PLP1 functions to permeabilize the PVM and the host plasma membrane ahead of PV rupture and egress, thereby resulting in ion fluxes including exposing the parasite to environmental $Ca^{2+}$ to amplify $Ca^{2+}$ signaling. This amplification serves to further enhance microneme secretion (including the release of more PLP1), motility, and acidification of the PV as a feed forward mechanism to mediate exit from host cells. In this scheme, PLP1 is potentially both a facilitator and a benefactor of PV acidification, with these elements occurring in succession prior to and during egress, respectively.

Exchangers, transporters, and $H^+$-ATPases on the parasite plasma membrane are plausible candidates to mediate the flux of $H^+$ necessary for the observed decrease in PV pH. Despite the delayed egress defects observed in parasites lacking VHA1, NHE1, or NHE3 [22,23, 25], we found that such mutants showed similar PV pH changes as parental or control parasites. Knockdown of an uncharacterized sodium/H+ exchanger, NHE4, also showed no effect on PV pH.

Several studies have identified formate-nitrite transporters (FNTs) as transporters of monocarboxylate metabolites, such as formate and lactate [30–32]. FNT transport of substrates depends on the cotransport of $H^+$, thus export of lactate or other substrates is intrinsically

linked to acidification of the environment. Previous work suggested that FNT1 is the major FNT in tachyzoites [30–32]. To determine the role of FNT1 and potentially FNT2 in acidification of the PV during egress, we endogenously tagged FNT1 and FNT2 with an mNG-AIDTy cassette. Individual knockdowns of FNT1 and FNT2 had no effect on the drop in pH, but lactate and pyruvate release assays suggest that the addition of the bulky mNG-AID tag on these transporters may have affected their functions. Another possible explanation is the previously reported auxin-independent degradation of AID-tagged proteins in a broad range of systems [39–41]. Since antibodies to FNT1 or FNT2 are not available to detect endogenous protein levels, we cannot determine whether these tagged strains are downregulated in the absence of IAA. Regardless, taken together our findings suggest that FNT1 and FNT2 both contribute to the release of lactate and pyruvate (and cotransport of H$^+$) during egress, but that other transporters can also export H$^+$ for residual acidification of the PV in the absence of FNT1 and FNT2.

It was unexpected that the *iFNT2* parasites showed reduced lactate and pyruvate release given the recent finding that FNT2 contributes little to lactate transport in WT extracellular parasites [32]. One potential explanation is that FNT2 plays a more prominent role in export of lactate, as tested in our study, versus uptake of lactate as examined in Zeng et al [32]. A common finding between the two studies is that the lack of expression of all three FNTs had no effect on replication of *T. gondii* under *in vitro* culture conditions.

We did not pursue in great depth the link between Ca$^{2+}$ signaling and activation of glycolysis because it was beyond the scope of the study. Nevertheless, we noted that elevation of parasite intracellular Ca$^{2+}$ coincided with acidification of the PV, which aligns with a potential role for Ca$^{2+}$ in activating glycolysis with concurrent excretion of lactate and pyruvate for acidification of the PV. Accordingly, our observation that zaprinast treatment markedly enhances export of pyruvate in an FNT dependent manner warrants further attention. That zaprinast augments release of pyruvate but not lactate implies that the fraction of lactate made under inducing conditions does not change but that increased production of pyruvate is managed by exporting it in a manner that is partly dependent on FNT1 and FNT2. In other systems Ca$^{2+}$ is also known to activate mitochondrial production of ATP through oxidative phosphorylation [42]. If the activation of Ca$^{2+}$ signaling triggers ATP production from glycolysis and mitochondrial respiration, this presumably serves to meet the energy intensive demands of gliding motility. Confirming this would reveal an additional role for Ca$^{2+}$ signaling during egress on top of its critical contributions to activating microneme secretion and motility. However, other explanations for zaprinast induced release of pyruvate are plausible. In addition to its action on PKG, zaprinast was also shown to inhibit mitochondrial pyruvate carrier activity, resulting in an accumulation and release of pyruvate and aspartate in mouse retina [43]. It is thus also possible that increased release of pyruvate from parasites is influenced by zaprinast inhibition of mitochondrial pyruvate carrier. We also cannot discount the possibility that zaprinast inhibition of host mitochondrial pyruvate carrier elevates pyruvate in infected host cells for diffusion into the PV. Additional studies are required to determine the extent to which Ca$^{2+}$ signaling influences glycolytic flux.

We present conclusive evidence for a drop in pH in the PV during both induced and natural egress, and we propose a model of the molecular mechanism by which this could occur (**Fig 8**). In WT parasites, an egress signal increases parasite cytosolic Ca$^{2+}$ levels, leading to the secretion of PLP1 from the micronemes and permeabilization of the PVM and possibly the host plasma membrane. Influx of Ca$^{2+}$ from the medium into the host cell is followed by uptake by the parasite to potentiate Ca$^{2+}$ signaling to release more PLP1 and activate gliding motility for egress. The increase in Ca$^{2+}$ could also trigger the net release of H$^+$, coupled with lactate, pyruvate, or other anions, to acidify the PV and create an environment more amenable

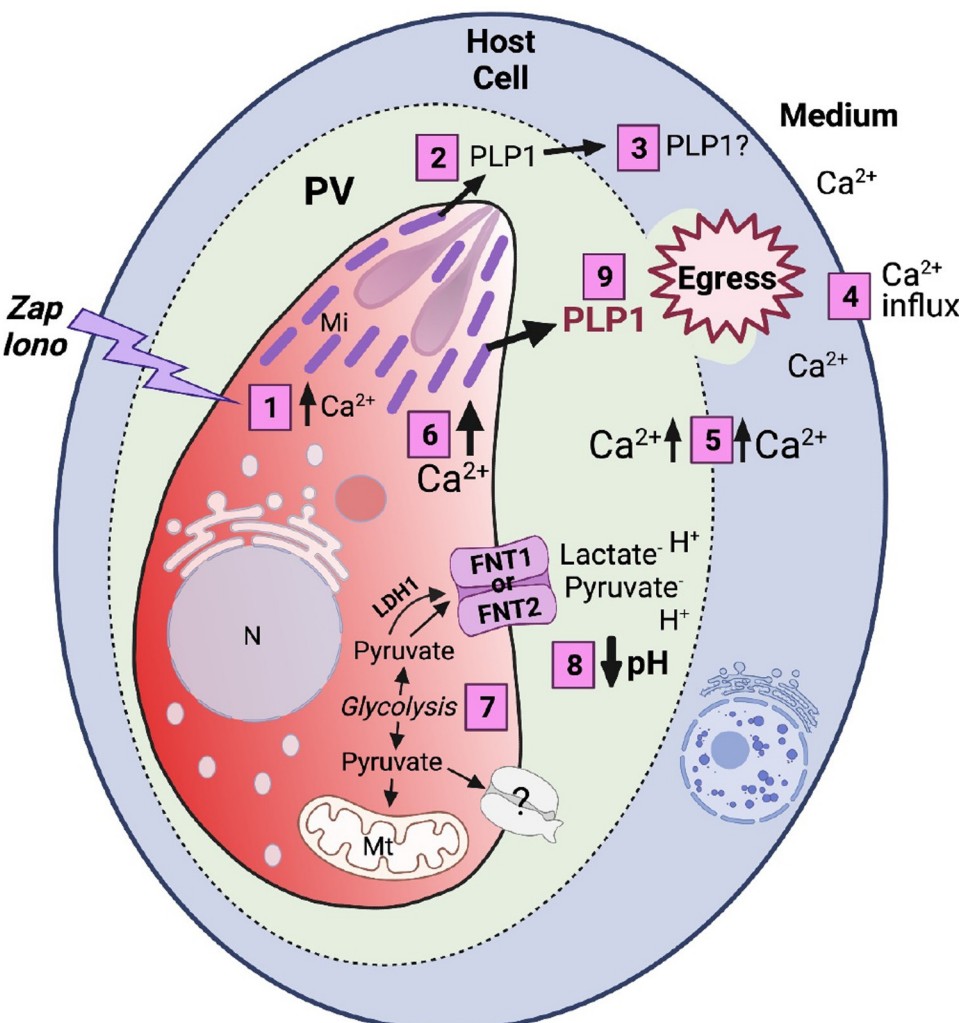

**Fig 8. Model for pH acidification.** Tachyzoites residing inside a parasitophorous vacuole (PV) within a host cell receive signal(s) that increase $Ca^{2+}$ in the parasite (1), leading to secretion of PLP1 from the micronemes into the PV (2) to form pores in the PV membrane and possibly host cell membrane. This allows an influx of ions such as $Ca^{2+}$ (4) into the host cell, into the PV (5), and potentiates another increase in the parasite (6), which results in greater microneme secretion and gliding motility (9). The products of glycolysis, lactate and pyruvate, are released into the PV space via formate-nitrite transporters (FNT1 and FNT2) (7). These products, in addition to other molecules via other unidentified transporters, reduce the PV pH prior to egress (8). N, nucleus; Mt, mitochondrion; Mi, micronemes. Fig 8 was created with BioRender.

to PLP1 cytolytic activity. FNT1 and FNT2 together contribute to the transport of the components for acidification, but clearly other transporters and exchangers can also function in this capacity. One potential candidate is the aquaporin water channel, AQ1, previously localized to the PLV/VAC compartment in tachyzoites [44]. A second aquaporin (AQ2) is also present in the *T. gondii* genome (toxodb.org). Several aquaporins have been shown to transport lactate (reviewed in [45]). A recent subcellular proteomics study suggested the localization of AQ1 in the Golgi or the plasma membrane (PM), depending on the prediction program used for the analysis [26]. If a fraction of AQ1 (or AQ2) is localized to the PM, it could play a role in the export of lactate. Another possibility is that the transporters and exchangers tested in this study act in concert. Amiloride is a potent inhibitor of $Na^+/H^+$ exchangers [46,47]. RH-RatpH

vacuoles treated with or without amiloride followed by induction with either zaprinast or ionomycin showed a significant abatement of a drop in pH in the PV (**S10 Fig**)**.** Although disruption of NHE1, NHE3, and NHE4 individually did not affect PV acidification, it is possible that ablation of two or more exchangers could have a greater effect. Additionally, the presence of multiple putative transporters, exchangers, P-type ATPases and proton ATPases in the *T. gondii* genome implies a pool of potential contributors to the drop in pH during egress. Identifying additional players could be the focus of future work. While we have shown that a pH drop might not be essential for parasites to leave the PV and host cell *in vitro*, this acidification could have a more critical role *in vivo* or in different stages of the parasite life cycle.

## Materials and methods

### Plasmids and parasite strains

Ratiometric pHluorin (RatpH) [16] was codon-optimized and chemically synthesized by GenScript Inc, as described previously [9]. All primers used are found in **Table 2**.

The codon optimized pHluorin was amplified with primer P30_RatpH.F and Ratph.R and Gibson cloned into the *Avr*II restriction site of the P30_GFP_GPI plasmid [48], replacing the GFP. The RatpH plasmid was co-transfected with a DHFR-TS selectable plasmid [49] into RH parasites and selected with 1 μM pyrimethamine. Individual clones were isolated by limited dilution in 96-well plates, and one validated clone was selected and termed RH-RatpH. RH-RatpH was transfected with a red genetically encoded $Ca^{2+}$ indicator (jRGECO1a in pCTH3, kindly provided by Dr. Silvia Moreno, University of Georgia) and an individual clone was isolated following selection with 20 μM chloramphenicol. These parasites are termed RH-RatpH-RGECO.

All mutant strains not stably expressing RatpH were transfected with 75–100 μg of RatpH and transiently transfected vacuoles were used in live imaging and pH experiments 28–30 h post-transfection.

Ratiometric pHluorin (Addgene, VV064: 1xCox8—ratiometric pHluorin in fck) optimized for expression in mammalian cells was PCR amplified and cloned in the EF-1α promoter driven pBud.CE4.1 plasmid in the *Kpn*I and *Bgl*II sites. For HeLa transfections, ~80% confluent HeLa cells in an 8-well Ibidi slide was transfected with 500ng of pBud.pHluorin with 1.5μl of Fugene (Promega) and 50μl of Opti-MEM. Cells were infected with RH-RatpH ~40hrs post-transfection and imaged ~28-30hrs post-infection.

FNT1 and FNT2 CRISPR gRNA constructs were generated for C-terminal tagging using the NEB Q5 Site Directed Mutagenesis kit according to Brown et al. [50]. Two FNT1 gRNAs (FNT1.gRNA1.Q5.F and FNT1.gRNA2.Q5.F) and two FNT2 gRNAs (FNT2.gRNA1.Q5.F and FNT2.gRNA2.Q5.F) with amplified with Cas9-ble.Q5.R were introduced into the Cas9-ble vector [51]. Repair templates containing 40 bp of homology at the 5' end and at the 3' end to FNT1 or FNT2 and flanking the mNeonGreen.mAID.Ty (mNG-AIDTy) cassette were amplified [52]. For transfection, 20 μg of each repair template along with 10 μg of each of the two gRNA constructs were transfected into RH*ΔhxgprtΔku80*:TIR parasites [27]. Upon lysis 2 days post-transfection, extracellular parasites were selected with a final concentration of 50 μg/ml phleomycin in DMEM for 6 h at 37°C and 5% $CO_2$, washed with DMEM, and placed back into a T25 containing HFF cells. Parasites were then cloned into 96-well plates and tested for the integration of the mNG-AIDTy cassette by PCR with primers FNT1.integ.F and Ty.Tag.R for FNT1 and FNT2.integ.F and Ty.Tag.R for FNT2

FNT2 was tagged at the C-terminal end in *iFNT1* parasites using the same CRISPR-Cas9 gRNA constructs as used for introducing the mNG-AIDTy cassette. The repair template was amplified from the pLinker.6xHA.HXGPRT.LoxP plasmid [53] and consisted of the 6xHA and HXPGRT selectable marker. Twenty micrograms of the repair template was transfected

**Table 2. Primers used in this study.**

| | Primer Name | Primer Sequence (5'…..3') |
| --- | --- | --- |
| Ratiometric pHluorin cloning for *T.gondii* | P30_RatpH.F<br>RatpH.R | -TTGCTGACCCTGCAGCATGCATGCTAGCTTTGTATAGTTC<br>-GCTGACCCTGCAGCATGCATGCTAGCttaTTTGTATAG |
| Ratiometric pHluorin cloning in pBud.CE4.1 | pBud.KpnI.RatpH.F<br>pBud.BglII.R | - cgcttcgaaggtacCAAAatgagcagtaaaggagaag<br>- ggcccagccggccagatcTCatttgtatagttcatc |
| FNT1 gRNA1<br>C-term tagging | FNT1.gRNA1.Q5.F | -atgaAGGAATGCGCCTGAAAGGTTTTAGAGCTAGAAATAGC |
| FNT1 gRNA2<br>C-term tagging | FNT1.gRNA2.Q5.F | -aCGCCCACAGCgacaagctgaGTTTTAGAGCTAGAAATAGC |
| FNT1 mNG-AIDTy Repair Template | FNT1.mAID.link.F<br>FNT1.mAID1.Ty.R | - CGGAGGGACGGGTGCCGCCGCCTCGCCCACAGCgacaagcAGCG<br>GGAGTGAGACGCCGGG<br>-GCGTTTTTTCTGGAAAACGCATATCCATCCTCACACGAATCTTAAT<br>CGAGCGGGTCCTGGTTCGTG |
| FNT2 gRNA1<br>C-term tagging | FNT2.gRNA1.Q5.F | -aCGGAAGTGCACACTCATGCCGTTTTAGAGCTAGAAATAGC |
| FNT2 gRNA2<br>C-term tagging | FNT2.gRNA2.Q5.F | -ACAATGTCAGCAGACTGCGCGTTTTAGAGCTAGAAATAGC |
| FNT2 mNG-AIDTy Repair Template | FNT2.mAID.link.F<br>FNT2.mAID2.Ty.R | - GGTTCCTTCTCAGACTGCTGAATCGGTGGCCCAGCAagtgAGCGG<br>GAGTGAGACGCCGGG<br>- CTCTACCACTTCTAAACTACACGCACTCGAACAGAATAATTCTTAA<br>TCGAGCGGGTCCTGGTTCGTG |
| FNT2 HA tagging<br>Repair Template | FNT2.HR1-L<br>FNT2.HR2-T | -TTATGGTTCCTTCTCAGACTGCTGAATCGGTGGCCCAGCAagtggctagc<br>AAGGGCTCGG<br>- CTCTACCACTTCTAAACTACACGCACTCGAACAGAATAATTCAATA<br>CGACTCACTATAGG |
| FNT2 gRNA1<br>N-term | FNT2.Nerm.gRNA1.F | -ATCCGGGGCTAGTCTTTCTGgtTTTAGAGCTAGAAATAGC |
| FNT2 gRNA2<br>N-term | FNT2.Nerm.gRNA2.F | -TCTCGACCAATCTGGGAACGgtTTTAGAGCTAGAAATAGC |
| HXGPRT gRNA1<br>For KO of FNT2 | HXG.gRNA1.F | AAGTTATCTGCAGTCTAGAGGTTTTAGAGCTAGAAATAGC |
| HXGPRT gRNA2<br>For KO of FNT2 | HXG.gRNA2.F | GAATTGGAGCTCCACCGCGGGTTTTAGAGCTAGAAATAGC |
| FNT2 KO with DHFR-TS Repair Template | FNT2.5'dhfrKO.F<br>FNT2.3'dhfrKO.R | -CTCCGTACACAGGCTTGACGGACGAATCGATCCGGGGCTAcagca<br>cgaaaccttgcattc<br>-CTACCACTTCTAAACTACACGCACTCGAACAGAATAATTCctgcaaGt<br>gcatagaaggaa |
| Absence of FNT2 gene | FNT2.256.F<br>FNT2.2415.R | -ATACCGTGCTTTTCCCAGTG<br>-AGATTGGGTTGCTACCATGC |
| Reverse Primer for gRNA Q5 | Cas9-ble.Q5.R | |
| FNT3 gRNA1<br>N-term tagging | FNT3.Nterm.gRNA1.F | ggtgcttgcggccaGTCCTGgtTTTAGAGCTAGAAATAGC |
| FNT3 gRNA2<br>N-term tagging | FNT3.Nterm.gRNA2.F | TATCGGAAAGTGATCGAATAgtTTTAGAGCTAGAAATAGC |
| FNT3 gRNA1<br>C-term tagging | FNT3.Cterm.gRNA1.F | aTGAAGAAACTCGCGTTGCTGgtTTTAGAGCTAGAAATAGC |
| FNT3 gRNA2<br>C-term tagging | FNT3.Cterm.gRNA2.F | AAGGAGGAGGAAGACGGCGGgtTTTAGAGCTAGAAATAGC |
| FNT3 KO with Bleomycin Repair Template | FNT3.GRA1_5'.ble.F<br>FNT3_3'.LDH.ble.R | -GTTTTCTTTTCCCTGGTCTTCTCTCCTGTTTCCTCTCCTCAAGCTT<br>CGAAGGCTGTAGTA<br>- CTTTCTTTTTCTTCGCATTCTCGCCTTCGCCGCGGCTCTCggaactac<br>ggtgtttgttcc |
| LDH1 N-term gRNA | LDH.Nterm.gRNA.F | atggcacccgcactTGTGCAGgtTTTAGAGCTAGAAATAGC |
| LDH1 C-term gRNA1 | LDH.Cterm.gRNA1.F | aaGCGTTGGCAAAACAGGAGgtTTTAGAGCTAGAAATAGC |
| LDH1 C-term gRNA2 | LDH.Cterm.gRNA2.F | GGAATGCCACTTTACTGCGCgtTTTAGAGCTAGAAATAGC |

*(Continued)*

**Table 2.** (Continued)

| | Primer Name | Primer Sequence (5'. ...3') |
|---|---|---|
| LDH1 KO with Bleomycin Repair Template | LDH.GRA1_5'.ble.F<br>GRA1_3'.LDH.ble.R | - GCAGACAACATCTGGCAGCCTCCCGCTCATTTTTAGTCAGAAGCT<br>TCGAAGGCTGTAGTA<br>- TTTCGCTTCCGTTGCAAACGCGTGTATAAATCATGGGCCCggaacta<br>cggtgtttgttcc |
| Integration of Ty tag in FNT1 locus | FNT1.integ.F (1a)<br>TyTag.R (1b) | ACATGTTCGGTCTCGAGGAT<br>CGGGTCCTGGTTCGTGTGGACCTC |
| Integration of Ty tag in FNT2 locus | FNT2.integ.F (2a)<br>TyTag.R (2b) | GAAGCAACTGGGCTTACGAC<br>CGGGTCCTGGTTCGTGTGGACCTC |
| Integration of HA tag in FNT2 locus | FNT2.256.F (c)<br>HAtag.R (d) | ATACCGTGCTTTTCCCAGTG<br>CCGGGTTAGGCATAATCTGG |
| Integration of dhfr-ts in FNT2 locus | FNT2.-568.F (e)<br>dhfr-ts.896.R (f) | CCCACTCACAAGTCCGGTTA<br>GAATCCTTGTACTCTTCCTCCAGAAGG |
| Integration of ble in FNT3 locus | FNT3.-98.F (g)<br>ble.55.R (h) | TCTCTCTTTTCGTCTCGCTTC<br>CGAAGCCCAACCTTTCATAG |
| Integration of ble in LDH1 locus | ble.22.F (i)<br>LDH.+275.R (j) | GCCATCACGAGATTTCGATT<br>CGAAGCCCAACCTTTCATAG |
| FNT3 qRT-PCR Set 1 | FNT3.qPCR.F1<br>FNT3.qPCR.R1 | CAGTGGCGGACTTCTTCTTT<br>GACGCTCCTTCTTCTTCTTCTC |
| FNT3 qRT-PCR Set 2 | FNT3.qPCR.F2<br>FNT3.qPCR.R2 | GCAGCCTACTTTCTCTCTTACC<br>GGCAGAAGCACTGGAGAAA |
| Actin qRT-PCR Set | actin.qPCR.F<br>actin.qPCR.R | GGGACGACATGGAGAAAATC<br>AGAAAGAACGGCCTGGATAG |

into *iFNT1* parasites, followed by selection with mycophenolic acid/xanthine. Integration of the 6xHA tag into the FNT2 locus was detected by PCR with primers FNT2.256.F and HAtag. R. Individual clones were isolated by limiting dilution, resulting in the strain *FNT2*-HA.

FNT2 knockout: Two gRNAs targeting the N-terminal end of *FNT2* (FNT2.Nerm.gRNA1.F and FNT2.Nerm.gRNA2.F) and two gRNAs against the 3' end of the 6xHA.HXGPRT cassette (HXG.gRNA1.F and HXG.gRNA2.F) were generated in the Cas9-ble vector. A repair template to knock out *FNT2* with the DHFR-TS selectable marker was amplified with primers FNT2.5'dhfrKO.F and FNT2.3'dhfrKO.R and transfected with all four gRNAs (two for FNT2 and two for HXGPRT). Transfected parasites were selected with 1 μM pyrimethamine and individual clones were isolated and tested for the integration of the DHFR-TS selectable marker into the *FNT2* locus with primers FNT2.-568.F and dhfr-ts.896.R and for the absence of the FNT2 gene with FNT2.256.F and FNT2.2415.R. Individual clones were isolated by limiting dilution, resulting in the strain *iFNT1Δfnt2*.

An LDH1 knockout strain was generated with an N-terminal gRNA (LDH.Nterm.gRNA.F) and two C-terminal gRNAs (LDH.Cterm.gRNA1.F and LDH.Cterm.gRNA.2.F) and a repair template using LDH.GRA1_5'ble.F and GRA1_3'LDH.ble.R to amplify the bleomycin selectable cassette. The repair template and gRNAs-Cas9-ble were transfected into *iFNT1Δfnt2* and selected as described above, resulted in the strain *iFNT1Δfnt2Δldh1*.

The *iFNT1Δfnt2Δfnt3* strain was generated in a similar method with N-terminal gRNAs (FNT3.Nterm.gRNA1.F and FNT3.Nterm.gRNA2.F) and C-terminal gRNAs (FNT3.Cterm. gRNA1.F and FNT3.Cterm.gRNA2.F) and repair template amplification with FNT3.GRA1_5'. ble.F and FNT3_3'.LDH.ble.R.

## Live imaging and induced or natural egress

HFF monolayers were plated onto Ibidi 8-well chamber slides (#1.5 polymer coverslip) in phenolred-free DMEM/10% cosmic calf/20 mM HEPES and grown overnight. Parasites

expressing pHluorin were inoculated into the slides the next day and allowed to grow for 28–30 h prior to imaging. Wells were washed with Ringer's buffer to remove any traces of background observed with DMEM and placed in 150 μl of Ringer's. Slides were placed in a heating chamber on a Zeiss AxioObserver at 37˚C and 5% $CO_2$. Solutions to be used were also kept at 37˚C in a heating block.

Following equilibration of the slide to the temperature and environment of the chamber, the time course experiment was started, and 5 frames were taken as the baseline measurement. This was followed by the addition of 150 μl of Ringer's containing 2x inducer (400 μM zaprinast or 2 μM ionomycin) with tubing. For cytochalasin D or mycalolide B experiments, wells were incubated in 2 μM of cytochalasin D or 3 μM of mycalolide B in Ringer's for 2 min prior to addition of inducer. All vacuoles enumerated had at least 8 parasites per PV.

For natural egress, infected Ibidi slides were infected with RH-RatpH and allowed to develop at 37˚C and 5% $CO_2$. Videos were collected following ~48–52 h of incubation and one frame was taken every 10sec for 20min. A new vacuole was then observed due to photobleaching of the vacuole.

The settings for taking images were as follows: Binning at 2x2, gain of 2, 1 frame every 1.5–5 seconds (or as indicated on graphs), 150 ms exposures each of the 410 nm and 470 nm channels. Custom pHluorin filter cube (Chroma) with excitation 395–415 nm and emission 500–550 nm for the 410 nm measurement and Filter Set 38HE (Zeiss) with excitation 450–490 nm and emission 500–550 nm for the 470 nm measurement were used. For parasites that also express RGECO, an additional image using a filter cube with excitation 540–580 nm and emission at 593–668 nm were taken. Images were converted from Zeiss czi files and exported as tiff files.

## pH calibration

RatpH expressing parasites in Ibidi chamber slides were incubated with Ringer's buffer and 30 μM nigericin at pHs ranging from 5.5 to 8.5 and allowed to equilibrate for 3 min. At each pH, a 410 nm measurement and a 470 nm measurement were taken. These values were used to generate a ratio of 410/470 nm, which was plotted against the known pH to create a pH calibration curve in Prism. The resultant linear regression equation was then applied to the experimental ratios to extrapolate a pH value.

## pH measurements in vacuoles

Each timed experiment at each channel was imported into ImageJ (one video at 410 nm, one at 470 nm, and one at 560 nm), converted to 32-bit, and a StackReg Plugin was used to align images. An individual vacuole in the 410 nm channel was selected with the FreeHand Selector and marked as a region of interest with ROI Manager. The Measure Stack Plugin was used to give the measurement of the Raw Integrated Density. This ROI was then measured in the 470 nm and the 560 nm channels (if the strain tested expresses RGECO). A ratio was generated by dividing the 410 nm value by the 470 mm, and a pH value was extrapolated using the pH calibration curve and the associated linear regression equation. The pH values for each vacuole were graphed in Prism. To quantify the decrease in PV pH from the tracings we identified the lowest pH value and subtracted it from pH value that was the highest immediately prior to the descent. If no obvious descent was observed (i.e., as seen with some treatments and mutants described below) then we took the pH value from the time of PV rupture and subtracted it from the value measured 15 sec earlier. Since DMSO treated samples showed no clear descent and they lacked PV rupture as a reference point, we calculated the average time to PV rupture for ionomycin induction and used it as the reference point for measuring PV pH relative to the value measured 15 sec earlier.

In addition to a pH calibration curve, each experiment included a timed experiment using DMSO in place of an egress inducer. This provides a measurement of the changes in pH from photobleaching of one of the channels. If it appeared that the pH trace changed/dropped through the time course, then the experimental traces were normalized by fitting a linear regression curve of the slope of DMSO trace.

### Buffer composition pH measurements

Chamber slides inoculated with RH_RatpH parasites were grown in phenolred-free DMEM for 28–30 h as described above. Prior to imaging, chamber wells were rinsed with warm Ringer's buffer and then incubated with the test buffer for 5 min. Induced egress and timed experiments were then started as described above. All buffers used are found in **Table 1**.

### Host Ca$^{2+}$ measurements

HFF cells inoculated with RH-RatpH, RH-RatpH-RGECO, Δ*plp1*-RatpH, or Δ*plp1*-RatpH-RGECO were incubated with 5 μM Cal-590-AM (AAT Bioquest, Cat. # 20510) and 0.04% pluronic acid (Biotium, Cat. #59004) in Ringer's for 30 min at 37˚C, followed by a 10 min wash with Ringer's. Egress induction was performed as above with zaprinast. CytD-treated parasites were incubated with 2 μM CytD for 2 min prior to induction. HeLa-jRGECO1a cells were plated onto Ibidi slides and inoculated as described for HFF cells. For host cell calcium measurements, frames were taken at 2 s intervals for Cal-590 and pHluorin release (only 470 nm channel) or 2.5 s intervals for Cal-590 and ratiometric pH measurements (both 410 nm and 470 nm channels).

### FNT inhibitors

BH-296 and BH-388 were generously provided by Drs. Holger Erler and Eric Beitz (University of Kiel). Stock solutions of 10 mM were prepared in DMSO, frozen in 5 μl aliquots, and diluted to final working solutions of 1 μM, 5 μM, or 10 μM in Ringer's buffer.

### Auxin induced depletion of mAID-tagged proteins

A stock of 500 mM of indole-3-acetic acid (IAA) in 100% ethanol was diluted 1:1000 to knock-down expression of mAID-tagged proteins. Parasites were incubated in 500 μM IAA for 1 h, 2 h, and 6 h prior to isolation of the lysates for western blot and probing with RbαTy antibody. Similarly, intracellular parasites in chamber slides were incubated with 500 μM of IAA and observed live for the loss of mNG or fixed and IFAs were performed using RbαTy antibody.

### Immunofluorescence microscopy

HFF monolayers on glass chamber slides were infected with parasite strains and fixed with 4% paraformaldehyde. Slides were permeabilized with 0.1% Triton X-100, blocked with 5% FBS and 5% BSA, and probed with primary antibody in Wash Buffer (WB, PBS/1% FBS). Slides were then washed with WB and incubated with secondary antibody in WB. Following washes, slides were mounted with Mowiol and viewed on a Zeiss AxioObserver.

RbαTy 1:1000, RbαGAP45 1:1000, MsαHA 1:1000, MsαIMC 1:500,

### Western blotting

Parasites were filter-purified, chased with cold PBS, pelleted at 1,000g at 4˚C, washed with cold PBS, and centrifuged again. Parasite pellets were resuspended in RIPA buffer (50 mM Tris-HCl [pH 7.5], 1% NP-40, 0.5% sodium deoxycholate, and 0.1% SDS, 150 mM NaCl) and incubated at room temperature for 10 min. Lysates were centrifuged at 15,000g at 4˚C for 10 min

and the supernatant was removed. Room temperature 4x NuPAGE LDS sample buffer was added to the supernatant, along with 2-beta-mercaptoethanol to a final concentration of 2%. Lysates were separated on SDS-PAGE gels and semi-dry blotted onto PVDF or nitrocellulose membranes. Membranes were blocked with 5% milk in PBS-Tween, incubated with antibodies diluted in 1.25% milk in PBS-Tween, and visualized with West Pico ECL substrate (Thermo Scientific).

## Statistical analyses

All statistical analyses were performed in GraphPad Prism. For each set of data, outliers were identified and removed using the ROUT method and an aggressive Q value of 0.1%. The data were then tested for normality and lognormality using the D'Agostino-Pearson test. If the data passed, a parametric One-way ANOVA with Tukey's multiple comparisons test was used. If any set of data failed the normality test, a non-parametric Kruskal-Wallis test was used.

Results were corroborated by a Mixed Effect Model generated from an RStudio script written by CSCAR (Consulting for Statistics, Computing, and Analytics Research) at the University of Michigan and considers random effects of the experiment.

## L-lactate and pyruvate detection assay

Parasites with and without IAA treatment for 48 h was purified through a 3 μM filter, chased with Ringer's buffer, and resuspended to $4x10^8$/ml. One hundred μl of each strain and treatment were placed into one well of a round-bottom 96-well plate and floated in a 37˚C water bath. To each well, 100 μl of prewarmed 400 μM zaprinast (2X) was added and incubated for 10 min. The plate was then moved on ice for 5 min, and then centrifuged at 4˚C for 10 min. After centrifugation, 175 μl of the supernatant was removed from each well, placed in another well, re-centrifuged, and 150 μl was then removed. For each lactate detection assay well, 25 μl of the supernatant was used. Lactate detection was performed following manufacturer's protocol, Promega Lactate-Glo (Cat. #J5021), in a white 96-well plate and read on a BioTek Synergy H1 plate reader, Luminescence endpoint, Gain 135.

For the pyruvate assay, 20 μl of supernatant was tested following the manufacturer's protocol, Cayman Chemical (Cat. #700470). Black plates were read on a BioTek Synergy H1 plate reader, with excitation at 530 nm and emission at 590 nm.

## Reverse transcriptase qPCR

RNA extraction of parasites was performed using Qiagen RNEasy columns, followed by cDNA synthesis with Invitrogen Superscript III First-Strand Synthesis. cDNA products were used with SYBR Green Mix using the following reaction conditions: 5 min at 95˚C, then 10 sec at 95˚, 30 sec at 60˚, 5 sec at 65˚C x 39 cycles. The fold change in relative expression was calculated using actin as a housekeeping gene for normalization.

## Supporting information

**S1 Fig. Expression of ratiometric pHluorin has no effect on the parasite response to egress induction or fitness.** A) Analysis of egress efficiency after ionomycin or zaprinast induction. B) Measurement of time to egress after ionomycin or zaprinast induction. C) Analysis of parasite fitness, as measure by a co-culture competition assay wherein RH (37% of the starting population) and RH-RatpH (63% of the population) were enumerated after passages 1, 2, and 5 and expressed as a percentage of the total population.
(TIF)

**S2 Fig. Time-lapse imaging of Ca$^{2+}$ (red traces) and pH (black traces) measuring Ca$^{2+}$ first followed by pH (solid lines) and switching the filter cube order to measure pH followed by Ca$^{2+}$ (dotted lines).** Green time points indicate PVM rupture.
(TIFF)

**S3 Fig. A) 410 nm image of HeLa cells transfected with RatpH.** Arrowhead indicates a transfected HeLa cell and arrow indicates a transfected and infected (with RH-RatpH) HeLa cell. B) Ratio image (410/470 nm) of A), dashed lines indicate the vacuoles. C) Time-course of ratio images following ionomycin induction; time post-induction is indicated in the upper left corner of each image. D,E) pH tracings of infected and uninfected-transfected host cells or vacuoles induced with DMSO or ionomycin (D) or zaprinast (E). Scale bar, 10 μm. Green data points indicate PV rupture.
(TIF)

**S4 Fig. PCRs to confirm knockouts of NHE1 and NHE3.** A) Primers designed to amplify a product only when a knockout of NHE1 has occurred (as used in (Arrizabalaga et al., 2004). B) Primers designed against the NHE3 gene detects NHE3 in WT but not Δ*nhe3* parasites.
(TIF)

**S5 Fig. NHE4-AIDTy localization and protein down-regulation.** A) TIR or NHE4-AIDTy parasites grown in chamber slides for 24 h were fixed and stained with anti-Ty and anti-GAP45 antibodies. B) NHE4-AIDTy parasites treated for 24 h with or without IAA were purified and immunoblotted with anti-Ty. Immunoblotting with anti-tubulin was used as a loading control. Units represent apparent molecular weight in kiloDaltons.
(TIF)

**S6 Fig. Knockdown and knock-out strategies.** A) Schematic diagram of the strategies used to endogenously tag or knock out genes, and the lineage. B) PCRs to detect integration of epitope tags or selectable markers as indicated by the primers (shown as letters) in A. Units represent size in kilobase pairs.
(TIF)

**S7 Fig. Effectiveness of knockdown with IAA treatment.** A) FNT1-AIDTy parasites were inoculated into Ibidi chamber slides and treated with or without IAA for the indicated times. The mNG fused to FNT1 was visualized with live imaging using a YFP filter cube. B) TIR and FNT1-AIDTy parasites were treated for the indicated times with or without IAA, filter purified, and lysates immunoblotted with anti-Ty. Arrow indicates FNT1-AIDTy and the asterisk indicates a non-specific band recognized by the Rb-Ty Ab. C) FNT2-HA tachyzoites were treated with and without IAA for 24 h and lysates immunoblotted for FNT2 (anti-HA), FNT1 (anti-Ty), and tubulin as a loading control. Units represent apparent molecular weight in kiloDaltons.
(TIF)

**S8 Fig. Reverse transcriptase qPCR to detect FNT3 mRNA expression.** Two separate primer sets were used. Data represent 5 biological replicates each with triplicates samples. Error bars are mean ± S.E.M.
(TIF)

**S9 Fig. Replication Assays.** Parasites were inoculated into 8-well chamber slides and allowed to replicate for 24 h prior to enumeration. A minimum of 250 PVs were counted. Data represent 3 biological replicates each with triplicate samples. Error bars are mean ± S.E.M.
(TIF)

**S10 Fig. Impairing Na⁺/H⁺ exchangers attenuates PV acidification.** RH-RatpH vacuoles incubated with or without amiloride. Data points represent changes in PV pH starting from baseline to a drop greater than 0.05 following induction with either ionomycin or zaprinast. (TIF)

**S1 Video. Time-lapse video of 410/470 nm images from the HFF cell infected with RH-RatpH shown in Fig 1D.** Parasites were induced with ionomycin and live imaged at intervals of 2.5 sec. (AVI)

**S2 Video. Time-lapse video of 410/470 nm images from the HFF cell infected with RH-RatpH shown in Fig 1G.** Imaging was done at ~50 h post-infection and natural egress was captured with live imaging at intervals of 10 sec. (AVI)

**S3 Video. Time-lapse imaging of intracellular Ca²⁺ levels (RGECO) in RH-RatpH-RGECO tachyzoites following zaprinast induction.** The graph to the right indicates the relative fluorescence units for Ca²⁺ corresponding to the time-lapse images on the left. (MP4)

**S4 Video. Time-lapse imaging of pH (pHluorin ratio) in an RH-RatpH-RGECO vacuole following zaprinast induction (same vacuole as in S3 Video).** The graph to the right indicates the relative pH values for pHluorin, corresponding to the time-lapse images on the left. (MP4)

**S5 Video. Time-lapse video of an HFF host cell preloaded with Cal-590-AM and infected with RH-RatpH-RGECO.** Video starts following zaprinast induction. Left panel, merged images; middle panel, RGECO images; left panel, 410 nm channel images showing pHluorin release. Time stamp on the merged image indicates the time for each channel. (AVI)

## Acknowledgments

We thank AJ Stasic, Stephen Vella, and Silvia Moreno for the *iΔvha1* parasites, RGECO plasmid and HeLa-jRGECO1a cells; Sebastian Lourido for the mNG-AIDTy plasmid; Gustavo Arrizabalaga for the Δ*nhe1* and Δ*nhe3* parasites; and Holger Erler and Eric Beitz for the FNT inhibitors and helpful suggestions. We appreciate the Consulting for Statistics, Computing, and Analytics Research (CSCAR) center at the University of Michigan. We thank Aric Schultz, Marijo Roiko, and Alfredo Guerra for critically reading the manuscript and members of the Carruthers lab for helpful discussions. Aric Schultz also helped to generate the videos in S3 Fig and S4 Fig.

## Author Contributions

**Conceptualization:** My-Hang Huynh, Vern B. Carruthers.

**Data curation:** My-Hang Huynh.

**Formal analysis:** My-Hang Huynh.

**Funding acquisition:** Vern B. Carruthers.

**Investigation:** My-Hang Huynh.

**Methodology:** My-Hang Huynh.

**Project administration:** My-Hang Huynh, Vern B. Carruthers.

**Resources:** My-Hang Huynh, Vern B. Carruthers.

**Supervision:** Vern B. Carruthers.

**Validation:** My-Hang Huynh.

**Visualization:** My-Hang Huynh.

**Writing – original draft:** My-Hang Huynh.

**Writing – review & editing:** My-Hang Huynh, Vern B. Carruthers.

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
