## [Decision Letter · Decision Letter 0]

30 Dec 2021

Dear Dr. Carruthers,

Thank you very much for submitting your manuscript "Toxoplasma gondii excretion of glycolytic products is associated with acidification of the parasitophorous vacuole during parasite egress" for consideration at PLOS Pathogens. As with all papers reviewed by the journal, your manuscript was reviewed by members of the editorial board and by several independent reviewers. In light of the reviews (below this email), we would like to invite the resubmission of a significantly-revised version that takes into account the reviewers' comments.

I am returning your manuscript with three reviews. After reading the reviews and looking at the manuscript, I recommend Major Revision based on the critiques. I am sorry I cannot be more positive at the moment. However, we are looking forward to receiving your revision addressing the critiques, if you so wish to do so. Note that we may send your paper back to the reviewers upon resubmission.

I would like to point out a special concern regarding the strength of the conclusions on the basis of the data presented and that progressive insight into the underlying mechanism is still premature. In addition, for the plp1ko experiments controls are not presented and it will need to be addressed to strengthen your claims and conclusions. Please pay particular attention to the following reviewer suggestions and give them due consideration.

We cannot make any decision about publication until we have seen the revised manuscript and your response to the reviewers' comments. Your revised manuscript is also likely to be sent to reviewers for further evaluation.

Sincerely,

Maria Belen Cassera, PhD

Guest Editor

PLOS Pathogens

Kami Kim

Section Editor

PLOS Pathogens

Kasturi Haldar

Editor-in-Chief

PLOS Pathogens

orcid.org/0000-0001-5065-158X

Michael Malim

Editor-in-Chief

PLOS Pathogens

orcid.org/0000-0002-7699-2064

I am returning your manuscript with three reviews. After reading the reviews and looking at the manuscript, I recommend Major Revision based on the critiques. I am sorry I cannot be more positive at the moment. However, we are looking forward to receiving your revision addressing the critiques, if you so wish to do so. Note that we may send your paper back to the reviewers upon resubmission.

I would like to point out a special concern regarding the strength of the conclusions on the basis of the data presented and that progressive insight into the underlying mechanism is still premature. In addition, for the plp1ko experiments controls are not presented and it will need to be addressed to strengthen your claims and conclusions. Please pay particular attention to the following reviewer suggestions and give them due consideration.

Reviewer's Responses to Questions

**Part I - Summary**

Reviewer #1: This historically narrated study reports elegant new reporters and techniques, with many new mutant parasites lines and experiments performed, however, only the last figure displays a modest set of data indirectly correlating the release of glycolysis products with vacuole acidification. Despite the large amount of work, progressive insight into the underlying mechanism is not well developed; although the presented data toward such a connection is intriguing, it presents many more loose observation points than lines that can connect them. Furthermore, data presentation and application of statistics are not consistent and not always correctly interpreted (notably, the details regarding measurement and interpretation of CytD/MycD use is not well presented). With that said, the imaging is very well done and tools are beautiful, which makes this more a tool development report than a major biological advance, or at least, premature.

Reviewer #2: In this study from Huynh and Carruthers, the authors attempt to address the contribution of acidification to parasite egress. Following up on their earlier work (Roiko et al, 2014), they use an improved pH-sensitive genetic reporter for better quantitation and interrogation of the event. They definitively confirm that PV acidification precedes both natural and stimulated egress, and that acidification is PLP1 dependent. These data are interesting, and strongly supported by their data. The authors then seek to identify the molecular basis and origin of acidification, providing a detailed analysis of ionic contribution, as well as proton pump candidates with an extensive genetic interrogation of a panel of potential molecular contributors.

Recommendation:

The experiments are carefully done (though the supp. figures need polishing), but some major concerns remain into the strength of their conclusions on the basis of the data they present, and the logic in the current narrative. These are detailed below. While these are major concerns, the contributions of the manuscript are sufficiently important and interesting that a revised manuscript should be considered for publication.

Reviewer #3: In this study, the authors investigated the acidification of the parasitophorous vacuole during egress of Toxoplasma gondii tachyzoites. For this purpose, they developed a novel assay based on genetically encoded ratiometric pHluorin (RatpH), which allows to monitor absolute variations in pH in live imaging experiments. Using these transgenic parasites, the authors document a drop of PV pH occurring immediately prior to induced and natural egress. When induced egress experiments were performed with plp1ko parasites expressing RatpH, an attenuation of the PV pH drop was observed, and a lack of acidification was even observed when motility was blocked (by cytD) in the plp1 mutant. The authors then tested several parasite transporters that could participate in acidification of the PV, using genetic approaches. They found that formate-nitrite transporters FNT1 and FNT2 contribute to acidification of the PV during egress, and that other unidentified factors also participate. Finally, based on analysis of extracellular tachyzoites, the authors conclude that release of lactate and pyruvate contribute to acidification of the PV during parasite egress.

Overall, the paper is very well written and clearly presented. The core of the study relies on a robust and elegant approach to measure pH using genetically encoded ratiometric pHluorin, allowing live imaging of PV pH during parasite induced or natural egress. The data provide definitive evidence that pH drops prior to egress. The study also includes a thorough analysis of genes potentially contributing to PV acidification, with single, double and triple mutants, using both conventional and conditional genetic strategies.

While a previous study from the same group showed that low pH promotes PLP1 secretion and activity (Roiko et al 2014), the authors now propose that PLP1 contributes to acidification of the PV, although the underlying mechanism remains unclear. However, this part of the work lacks important controls, as detailed below, to strengthen the conclusion that plp1 depletion attenuates the drop in PV pH. Furthermore, the relevance of using ESA from extracellular tachyzoites to measure lactate and pyruvate release is not obvious.

**Part II – Major Issues: Key Experiments Required for Acceptance**

Reviewer #1: (No Response)

Reviewer #2: Major comments:

• These genetically encoded reporters can buffer the event they are intended to report upon, and consequently affect cell physiology. This should be addressed by plaque assay for the reporter lines to show no gross different in parasite health as a consequence of expressing these reporters. Alternatively, a competition growth assay could be used to address this point.

• Line 248: ‘RH-RatpH-RGECO parasites displayed a rapid increase in cytosolic Ca2+ after ionomycin induction and well before the drop in PV pH and egress (Figure 3F and Supplemental Figure S2).’ While I agree with the author’s interpretation of the data, the major challenge of comparing signal outputs from different reporters is that the kinetics of signal detection and integration of the signal input for the reporter output are likely to be different. Which makes comparison of the two reporter signals inherently unfair. This should be more cautiously worded throughout, and this caveat noted in the discussion. Also, the speed at which the camera acquires the images (and channel order in which they are acquired) could affect the interpretation of the data, and possible result in an experimental artifact. To address this the authors should switch the order of channel acquisition to confirm that there is a genuine difference in the timing of the events.

• The experimental logic and rationale needs elaboration, as at the present time some sections are somewhat contradictory. My main point of confusion comes from the following: lines 225-229: ‘However, PVs of immobilized Δplp1-RatpH showed no decrease in pH and they failed to release pHluorin for the entire duration of the experiment (up to 20 min) in all 225 vacuoles observed (Figure 3E). Taken together, these findings suggest that PLP1 influences PV pH and that rupture of the PVM and possibly the host plasma membrane via PLP1 or motility is necessary for acidification of the PV’. My reading of this that microneme release of PLP1 is necessary for the process of acidification. The primary function of PLP1 is thought to be the formation of pores on the parasitophorous vacuole membrane, ultimately leading to parasite egress. This would indicate that acidification is likely host-cell dependent, taking place on the PVM. Following the PLP1 dataset (Fig. 3), the authors explore the molecular basis of acidification: Line 257: ‘Several plasma membrane proton transporters are not required for acidification of the PV’ – if acidification is PLP1-dependent (which they convincingly demonstrate), and PLP1 thought to be PVM localized following its release from the micronemes, why do the authors then explore parasite plasma membrane transporters? It is possible that this is simply an issue with the manuscript narrative structure – but as it’s currently written it is difficult to follow the logic of their work, and understand why this was done.

• Line 304: ‘We initially tested the concentrations of the inhibitors that were used in the previous study, but 10 µM, BH-296 blocked tachyzoite egress, precluding its use at this concentration’ – the basis for this is unclear. The authors previously use CytD and MycB to ‘paralyse’ parasite and remove variation that can be introduced into the microscopy studies by motile parasites. Treatment of these parasites with these paralyzing drugs prevents egress, but does not prevent the authors from assessing changes in pH. This discrepancy needs to be addressed.

• Line 416: ‘Parasites lacking all three FNTs (iFNT1Δfnt2Δfnt3+IAA) still showed residual release of pyruvate, implying the existence of other unidentified transporters capable of exporting pyruvate.’ As FNT1 is a conditional knockdown and not a clean knockout, the formal possibility remains that residual FNT1 is responsible – and that there is no need to imply the existence of other pyruvate transporters. This should be addressed in the text.

• Line 453-455: ‘Nevertheless, a drop in pH is not essential for exit from host cells, which is consistent with the event being regulated at multiple levels to ensure successful liberation.’ Do you mean is essential? Otherwise I’m not sure the logic of the sentence is consistent, or is confusing as currently written – it is not clear if ‘the event’ refers to the drop in pH or egress.

Reviewer #3: 1. The plp1 mutant experiments lack an important control in figure 3B/C, the non-induced condition (DMSO). How could the authors determine the difference in lowest regional PV pH for the plp1 mutant without having the non-induced condition (Line 216-217)? In fig 3F, the pH in plp1 mutant seems to be higher than RH. Could this explain the apparent lower pH drop in fig3B/C, where comparison was made with the non-induced RH parasite? In Fig3E, pH drops constantly in the plp1ko with cytD/MucB conditions, even before the addition of the inducer (ionomycin), suggesting a potential experimental issue.

2. The relevance of using ESA to analyze lactate/pyruvate transport is questionable (figure 6). ESA fractions likely represents components released upon induced exocytosis of apical organelles. FNTs or other transporters are expected to act at the parasite plasma membrane, therefore the relevance of the ESA experiments is not clear. This probably explains why the results are inconsistent with the Zeng et al study, which analyzed transport across the parasite plasma membrane. Because of this limitation, it is difficult to conclude on the contribution of lactate/pyruvate in acidification of the PV. The experiments using ldh1 mutants suggest that lactate production is not required for acidification of the PV. How then could lactate release contribute to PV acidification, as proposed by the authors?

**Part III – Minor Issues: Editorial and Data Presentation Modifications**

Reviewer #1: 1. Line 57 “PV pH during egress” should this be ‘chemically induced’ or ‘pharmacologically triggered’ egress, since a DMSO control is mentioned in the second half of the sentence, but it is not clear what this controls for.

2. Line 64 “lead an abatement” should this be ‘lead TO an abatement”?

3. Legend Fig 1D. please add that ionomycin was used as egress inducer

4. Fig 2; 3C. The horizontal line represents the mean, however, the Kruskal-Wallis + Dunn measures the ‘median’. This seems therefore to be a mismatch in graphical presentation and statistical analysis.

5. Fig 2B. The schematic says “zaprinast’ this should be ‘ionomycin’

6. Line 190: “Since CytD did not markedly affect PV pH…” the statistics for that is not shown, and there appears to be a shift to more aciditiy upon additions of CytD. Neither is a control of just CytD included.

7. Fig 3B. It is not indicated whether the line represent mean or median

8. Fig 3C. Assumingly, the Kruskal-Wallis + Dunn analysis was also applied here

9. Fig 3D, E. Both CytD and MycD display a gradually decreasing pH of about 0.5 units that equilibrates around 300 sec in (Fig 3E). As such, when the measurement was performed in Fig 2 in presence of CytD would have a major impact on experimental outcome (see #6 above as well). It was said 5 min in the M&M. Assuming the traces shown in Fig 3D, E are representative, it is surprising to see in Fig 2A, B that the average/median (?) pH drop in Fig 2A/B is less than 0.25. So somehow these observations do not add up to exactly the same thing in these two experiments.

10. Fig 4C-F. It would be helpful to also mark which comparisons are ‘not significant’, as (partly) done in 2B, 3C, 5A,E,F,H

11. Line 306 states “At both 1 uM and 5 uM, BH-296 reduced the magnitude of pH change during zaprinast-induced egress compared to RH-RatpH, …” However, in Fig 5A only 5 uM BH-296 is statistically significant, not 1 uM.

12. Text makes not reference to Fig 5E before mentioning Fig 5F

Reviewer #2: Minor comments:

• Figure 1E – it would be useful to include an indicator on the trace for the point of egress to be able to relate trace shifts with the physiological event.

• Line 232-233: ‘ratiometric pHluorin under a mammalian promoter’ – should be reworded for clarity – ‘under a mammalian promoter’ is meaningless.

• Fig. 3D and E: from the traces presented it appears that the starting pH of RH and ∆plp1 vacuoles are different – is this correct? Were vacuoles containing equal numbers of parasites compared? This should be discussed.

• The presentation of data in 5D and E is confusing – it would be better to indicate the DMSO lane are also RH parasites – this could be done with a second +/- line indicating the use of an egress inducer in that experimental condition, and relabeling DMSO to be RH.

• Line 251-254: regarding Fig. 3G – please provide details of statistical test used.

• Line 265: ‘generated in Stasic et al’ – the correct numerical reference has been missed.

• Figure 5C: the panel strain label has been incorrectly cropped off.

• Figure 7: ‘BCKDH, branched chain ketoacid dehydrogenase’ is included on the figure and in the legend, but not mentioned in the manuscript – please remove for simplicity or explain its inclusion.

• Figure S1: add scale bars

• Figure S2: add scale bars. also, for the pHluorin movie, the pH colored scale bar is misleading as the colors associated with each pH seem to change between time points. This means that a given color does not consistently represent a specific pH, and when viewing the data as a time series movie it makes it almost impossible to interpret the data real time. This should be corrected to have the colors used be consistent across the image series.

• Figure S3: add units for the numbers presented on the gels.

• Figure S4: add scale bars. Add units for the numbers presented on the gels.

• Figure S5b: add units for the numbers presented on the gels.

• Figure S6: add scale bars. Add units for the numbers presented on the gels.

Reviewer #3: -The number of PV analyzed to generate “representative tracings” should be specified in the figure legends (fig 3 and fig 4).

-The data show that FNT2 cannot be detected in iFNT2 or FNT2-HA parasites. Is there a compensatory overexpression of FNT2 in FNT1-depleted parasites? Could this explain why the combined depletion of both FNTs is effective in contrary to single knockdowns.

-Line 205-207: it I unclear how PLP1 and PVM breakage could contribute to ion flux to acidify the PV. Is it from the host cell? More generally, the model in fig7 would benefit from the addition of a temporal dimension specifying the sequence of events, as nicely done in the discussion.

-Fig2A label on the x axis should indicate ionomycin not zaprinast

-Figure S2 is empty

-Figure 3F/3G: how do the authors explain the delay in calcium in the plp1? Why is calcium level higher (as observed with pH) in the mutant?

-Figure 5F and 5H: it would be useful to show triple and double mutants in the same graph, to allow direct comparison.

PLOS authors have the option to publish the peer review history of their article (what does this mean?). If published, this will include your full peer review and any attached files.

Reviewer #1: No

Reviewer #2: No

Reviewer #3: No
---

## [Decision Letter · Decision Letter 1]

28 Mar 2022

Dear Dr. Carruthers,

We are pleased to inform you that your manuscript 'Toxoplasma gondii excretion of glycolytic products is associated with acidification of the parasitophorous vacuole during parasite egress' has been provisionally accepted for publication in PLOS Pathogens. As you go through the formatting changes, please note the comment from reviewer #1 to clarify the legend of Figure 4D: "Fig 4D; Legend mentions yellow and green marks, but the figure only has green marks. Please consolidate."

Best regards,

Maria Belen Cassera, PhD

Guest Editor

PLOS Pathogens

Kami Kim

Section Editor

PLOS Pathogens

Kasturi Haldar

Editor-in-Chief

PLOS Pathogens

orcid.org/0000-0001-5065-158X

Michael Malim

Editor-in-Chief

PLOS Pathogens

orcid.org/0000-0002-7699-2064

Reviewer Comments (if any, and for reference):

Reviewer's Responses to Questions

**Part I - Summary**

Reviewer #1: The authors have been very responsive to the concerns raised, and together with the new data, the manuscript quality has increased dramatically.

Reviewer #2: The authors have significantly revised their initial submission, making constructive use of the review and adding additional data that serves to strengthen their conclusions. The revised manuscript is excellent, and the efforts made by the authors to clarify their arguments and provide a useful working model for what is becoming an increasingly complex molecular model of egress much appreciated!

Reviewer #3: The authors made a great job in clarifying some aspects of the manuscript, including the methods to measure pH drop, and provide an updated and interesting model (Fig8) that will stimulate future research on how PLP1 affects calcium signaling in the host cell.

**Part II – Major Issues: Key Experiments Required for Acceptance**

Reviewer #1: (No Response)

Reviewer #2: Nothing required

Reviewer #3: (No Response)

**Part III – Minor Issues: Editorial and Data Presentation Modifications**

Reviewer #1: Fig 4D; Legend mentions yellow and green marks, but the figure only has green marks. Please consolidate.

Reviewer #2: Nothing required

Reviewer #3: (No Response)

PLOS authors have the option to publish the peer review history of their article (what does this mean?). If published, this will include your full peer review and any attached files.

Reviewer #1: No

Reviewer #2: **Yes: **Matthew A. Child

Reviewer #3: No

---

## [Editor Report · Acceptance letter]

29 Apr 2022

Dear Dr. Carruthers,

We are delighted to inform you that your manuscript, "Toxoplasma gondii excretion of glycolytic products is associated with acidification of the parasitophorous vacuole during parasite egress," has been formally accepted for publication in PLOS Pathogens.

Best regards,

Kasturi Haldar

Editor-in-Chief

PLOS Pathogens

orcid.org/0000-0001-5065-158X

Michael Malim

Editor-in-Chief

PLOS Pathogens

orcid.org/0000-0002-7699-2064